# Cytokine Networks in the Pathogenesis of Rheumatoid Arthritis

**DOI:** 10.3390/ijms222010922

**Published:** 2021-10-10

**Authors:** Naoki Kondo, Takeshi Kuroda, Daisuke Kobayashi

**Affiliations:** 1Division of Orthopedic Surgery, Department of Regenerative and Transplant Medicine, Graduate School of Medical and Dental Sciences, Niigata University, 1-757 Asahimachi-Dori, Chuo-ku, Niigata City 951-8510, Japan; naokikondo1214@gmail.com; 2Health Administration Center, Niigata University, 2-8050 Ikarashi, Nishi-ku, Niigata City 950-2181, Japan; 3Division of Clinical Nephrology and Rheumatology, Graduate School of Medical and Dental Sciences, Niigata University, 1-757 Asahimachi-Dori, Chuo-ku, Niigata City 951-8510, Japan; joifa602@hotmail.com

**Keywords:** rheumatoid arthritis, TNF-α, IL-6, IL-7, IL-17, IL-21, IL-23, GM-CSF, IL-1β, IL-18, IL-33, IL-2

## Abstract

Rheumatoid arthritis (RA) is an autoimmune disease characterized by chronic systemic inflammation causing progressive joint damage that can lead to lifelong disability. The pathogenesis of RA involves a complex network of various cytokines and cells that trigger synovial cell proliferation and cause damage to both cartilage and bone. Involvement of the cytokines tumor necrosis factor (TNF)-α and interleukin (IL)-6 is central to the pathogenesis of RA, but recent research has revealed that other cytokines such as IL-7, IL-17, IL-21, IL-23, granulocyte macrophage colony-stimulating factor (GM-CSF), IL-1β, IL-18, IL-33, and IL-2 also play a role. Clarification of RA pathology has led to the development of therapeutic agents such as biological disease-modifying anti-rheumatic drugs (DMARDs) and Janus kinase (JAK) inhibitors, and further details of the immunological background to RA are emerging. This review covers existing knowledge regarding the roles of cytokines, related immune cells and the immune system in RA, manipulation of which may offer the potential for even safer and more effective treatments in the future.

## 1. Introduction

Rheumatoid arthritis (RA) is a chronic, systemic inflammatory disease causing progressive joint damage that can lead to lifelong disability [1]. Although RA itself is not life-threatening, it may cause secondary amyloidosis, which carries a risk of organ failure and death [2]. This systemic disease is characterized by synovial inflammatory cell infiltration, synovial hyperplasia, angiogenesis and cartilage damage, which in turn can lead to bone destruction [3]. Various autoantibodies such as rheumatoid factor (RF) and anti-cyclic citrullinated peptide (CCP) antibody (ACPA) are important serological markers for not only RA diagnosis but also prediction of treatment responsiveness and progression of bone destruction [4]. The level of ACPA is elevated with high specificity in RA, and clinically it is widely used for diagnosis [5]. The production of ACPA is thought to be correlated with certain genetic background factors such as HLA-DR [6]. ACPA and RF form immune complexes with citrullinated proteins and activate macrophages, triggering the release of inflammatory cytokines such as tumor necrosis factor (TNF)-α and interleukin (IL)-6 [7]. This cytokine-mediated pathway is central to the pathogenesis of RA [8,9]. At the affected joint, infiltrating immune cells are associated with the release of a variety of cytokines, which are important mediators of cell differentiation, inflammation, immune pathology, and immune response [7]. As further details of RA pathogenesis have emerged, biological disease-modifying anti-rheumatic drugs (DMARDs) and Janus kinase (JAK) inhibitors have been developed as therapeutic agents, and the underlying immunological conditions are being further clarified [10]. It has become evident that numerous cytokines are associated with the pathophysiology of RA, and that some of them might become targets for future therapeutics. Here we review several of these cytokines, including TNF-α, IL-6, IL-7, IL-17, IL-21, IL-23, IL-1β, IL-18, IL-33, granulocyte macrophage colony-stimulating factor (GM-CSF), and IL-2, which are known to be active from the acute to the chronic stage of RA, and may have potential for therapeutic targeting [11,12]. The possible mechanism of RA pathogenesis is depicted in Figure 1.

## 2. TNF and RA

### 2.1. ACPA Stimulate Macrophages to Produce TNF

Primarily, the TNF-α precursor molecule is produced as a transmembrane protein (memTNF), which is then cleaved by a metalloproteinase, such as TNF-α converting enzyme (TACE), leading to the release of soluble TNF (sTNF) [13]. One of the major sources of TNF is monocyte-derived macrophages. There are two types of synovial macrophage in the mouse: one is an intrinsic macrophage in the joint synovium present from birth, and the other is an extrinsic macrophage derived from bone marrow. Intrinsic macrophages express anti-inflammatory cytokines such as IL-4 and IL-10, whereas extrinsic macrophages express proinflammatory cytokines such as IL-1β and TNF-α. Similar cells have also been identified in patients with RA [14]. Recent integrating single-cell transcriptomics and mass cytometry studies have revealed that B cells and T cells are also the major source of TNF in synovium affected by RA [13,15].

Macrophage activation is an important process in RA pathogenesis, and this can occur via the action of proinflammatory cytokines, immune complexes, and Toll-like receptor (TLR) agonists [16]. ACPA is a useful factor for classification of RA and is known to be a predictor of joint destruction [1,17]. Recent analysis has indicated that ACPA also contributes to the pathogenesis of RA. ACPA forms immune complexes that stimulate macrophages to produce TNF and other proinflammatory cytokines [18]. Interestingly, ACPAs with higher levels of somatic hypermutation have the ability to bind an expanded set of citrullinated antigens, thus inducing a higher amount of TNF [19].

### 2.2. Multidirectional Function of TNF in RA Pathogenesis

TNF is one of the key regulators of RA pathogenesis. Its expression is increased in RA patients, and overexpression of TNF causes autoimmune arthritis in transgenic animals [1,17]. TNF signaling is involved multidirectionally in the pathogenesis of RA. It activates endothelial cells and recruits proinflammatory cells such as synovial fibroblasts and macrophages, which release proinflammatory cytokines such as IL-6, IL-1β, and TNF [13,17,20,21]. It also controls the development of T helper (Th)1 and Th17 T cells, antibody production, and osteoclast differentiation [10,22,23]. Here we discuss recent findings related to the function of TNF in RA, focusing especially on its cellular sources and its contribution to the pathogenesis of RA, the function of TNFRs, regulatory T cell (Treg) development, and epigenomics.

Due to its multidirectional function and variety of cellular sources as mentioned above, much has remained unclear about the pathogenetic roles of TNF. Kruglov et al. performed a series of studies using mice with conditional TNF knockout [24]. These revealed that memTNF has a protective function against arthritis and suppresses autoreactive T cells. MemTNF was reported to preferentially bind to TNFR2. It was also shown that TNF produced by myeloid cells controls arthritis onset by regulating the activation of synovial fibroblasts, and that B cell-derived TNF regulates the severity of arthritis through induction of autoantibodies. On the other hand, T cell-derived TNF was shown to exert a protective action by regulating the development of autoreactive T cells.

### 2.3. TNFR1 and TNFR2

There are two known receptor isoforms: TNF receptors 1 (TNFR1) and TNFR2 [13]. TNFR1 is expressed ubiquitously, whereas TNFR2 is expressed on T cells, myeloid cells, and endothelial cells. Upon activation, a precisely controlled multistep ubiquitination process activates nuclear factor-kappa B (NF-κB) [13]. The intracellular signaling process is depicted in Figure 2.

When TNFR1 is knocked out in a model of collagen-induced arthritis (CIA), disease activity decreases, whereas TNFR2-knockout mice develop severe arthritis. Furthermore, both TNFR1 antagonist and TNFR2 agonist ameliorate the severity of arthritis in the CIA model [25,26,27,28]. Thus, the TNFR1 signal seems to activate inflammatory target genes, whereas the TNFR2 signal exerts a protective role by regulating the function of Treg cells. 

TNF affects the function and differentiation of Treg cells, which are essential for maintenance of immune homeostasis and prevention of autoimmunity, and whose differentiation is regulated by FoxP3 [29]. While signaling via TNFR1 promotes the pathogenesis of arthritis, TNFR2 signaling exerts protective functions [10,13]. Several lines of evidence suggest that interaction of TNF with TNFR2 promotes Treg function [30]. For example, the TNF-TNFR2 axis expands the number and function of mCD4+CD25+ Tregs. Human peripheral blood CD4+CD25+TNFR2+ cells have been shown to markedly inhibit the proliferation of, and cytokine production by, co-cultured T-responder cells, whereas CD4+CD25+TNFR2-cells do not. In a study using TNFR2-knockout mice, FoxP3 gene methylation in Treg cells was shown to be greater than in wild-type mice, and Treg cells differentiated into proinflammatory Th17-like cells. Thus, TNFR2 signaling appears to block methylation of the promoter region of FoxP3, thus maintaining FoxP3 transcription and preventing pathogenic conversion of Tregs to Th17-like cells [28]. Furthermore, transfer of TNFR2-expressing Treg cells has been shown to ameliorate inflammation in an experimental arthritis model [31]. On the other hand, TNFR1 deficiency increases Treg activity, suggesting that TNFR1 signaling mediates the disease exacerbation attributable to Tregs. TNF-TNFR1 signaling activates NF-κB, leading to transcription of protein phosphatase 1 (PP1), which dephosphorylates Ser418 of FoxP3. As dephosphorylated FoxP3 is unable to bind the target sequence, Treg differentiation is blocked [32]. Although the precise TNFR2 signaling mechanism in Tregs is under investigation, the increased proliferation and prolonged survival induced by TNFR2 activation is impaired in RelA-deficient Treg cells [33,34]. Recently, another ligand of TNFR2, progranulin (PGRN), has been identified. PGRN binds to TNFR2 with an affinity 600-fold stronger than to TNF [35]. TNFR2 stimulated by PGRN interacts with 14-3-3ε in the intracellular domain and activates the downstream cascade through phosphatidylinositol 3 kinase (PI3K)/Akt/mammalian target of rapamycin (mTOR), thus restricting NF-κB activation while simultaneously stimulating C/EBPβ activation [36]. Knockout of 14-3-3ε or PGRN exacerbates arthritis in the CIA model, with an increase of proinflammatory macrophages and a decrease of Treg cells in affected joints.

### 2.4. TNF in Epigenetics

During the course of RA, from the very early preclinical stage to the established disease stage, the DNA methylation status of RA synovial fibroblasts changes dynamically [37]. TNF controls gene expression by regulating both methylation and acetylation. It has been shown that hypomethylation is associated with increased gene expression, and that many genes involved in RA pathogenesis, including signal transducer and activator of transcription 3 (STAT3) and TNF receptor associated factor (TRAF)2, are hypomethylated in synovial fibroblasts derived from RA patients [38]. Cytokines such as TNF and IL-1β can inhibit the expression of DNA methyltransferase (DNMT), leading to reduced DNA methylation and increased pathogenetic gene expression [39]. Acetylation is another mechanism by which TNF can regulate gene expression. Histone deacetylases (HDAC) are a class of enzymes that remove acetyl groups from lysine on histone, allowing histones to wrap DNA more tightly. DNA expression is controlled through a balance between acetylation and deacetylation. HDAC5 is known to exert an anti-inflammatory role. TNF and IL-1β suppress HDAC5 and upregulate the production of inflammatory cytokines and chemokines [40]. Loh et al. investigated genome-wide changes in gene expression and chromatin remodeling induced by TNF in synovial fibroblasts and macrophages. They stimulated synovial fibroblasts from RA patients and human CD14+ monocyte-derived macrophages with TNF ex vivo, and identified 280 TNF-inducible arthritogenic genes, including IL-6, C-X-C motif chemokine ligand (CXCL)8, CXCL10, and matrix metalloproteinase (MMP)-19. These genes were expressed transiently in macrophages, but their expression was sustained in synovial fibroblasts [41]. The level of CXCL8 in synovial fluid or peripheral blood is higher in RA patients than in healthy controls. CXCL8 causes migration of immune cells to the joints, leading to joint destruction [42]. CXCL10 accelerates the migration of inflammatory cells in a CXC chemokine receptor (CXCR)3-dependent manner and induces receptor activator of NF-κB ligand (RANKL) expression in CD4+ T cells. Cxcl10–/– and Cxcr3–/– mice with collagen antibody-induced arthritis (CAIA) show milder joint destruction than the wild type [43], and the blood level of CXCL10 correlates with RA disease activity [44]. They further identified 80 genes that lost their reactivity with TNF when stimulated repeatedly in macrophages, but which retained their reactivity in synovial fibroblasts [41]. These genes in macrophages and synovial fibroblasts were subjected to Assay for Transposase-Accessible Chromatin using SEQuencing (ATAC-seq), which can identify open chromatin, and chromatin immunoprecipitation coupled with high-throughput sequencing (ChIP-seq) for Histone 3 lysine 27 acetylation (H3K27ac). The results indicated persistent TNF regulation of H3K27 acetylation and increased chromatin accessibility in the regulatory elements of arthritogenic genes in TNF-stimulated synovial fibroblasts [41]. Not only TNF, but also other cytokines contribute to epigenetic modification. When stimulated with a mixture of eight cytokines, including IL-1β, IFN-α, and IFN-γ, the genomic structure of synovial fibroblasts changes dramatically, with formation of chromatin loops that can be detected by high-throughput chromosome conformation capture (Hi-C) analysis. The transcription factors metal-regulatory transcription factor-1 (MTF1) and runt-related transcription factor-1 (RUNX1) could be key regulators of chromatin remodeling for expression of pathogenic molecules in fibroblasts [45]. Table 1 summarizes the RA-related cytokines, their signaling pathways, and clinical assessments described in this article.

### 2.5. Evaluation of Anti-TNF Agents and Challenges of the Future

Based on the rationale that TNF-α plays a central role in the regulation of RA-related molecules, anti-TNF drugs were the first biological agents to be introduced for treatment of RA, starting with infliximab, a chimeric anti-TNF-α monoclonal antibody, in 1999 [1,17]. Since then, several blocking agents have been approved with favorable clinical efficacy, and widely used in daily clinical practice [46]. Although TNF was originally identified as a factor that induced necrosis of tumor cells, recent meta-analysis and network meta-analysis has shown that treatment with biological DMARDs (bDMARDs) including TNF-inhibitor did not increase the risk for malignancies [47]. Benefits of bDMARDs (abatacept, adalimumab, anakinra, certolizumab pegol, etanercept, golimumab, infliximab, rituximab, tocilizumab) and tofacitinib have been demonstrated in network meta-analyses [48,49,50,51]. Furthermore, a recent individual patient data (IPD) network meta-analysis found only minor differences in benefits and harms among bDMARDs in patients who responded insufficiently to methotrexate (MTX) [52]. Although treatment of RA has been markedly improved by anti-TNF antibody, many issues remain unresolved. Current anti-TNF drugs not only inhibit pathogenetic TNF but also inhibit protective TNF derived from T cells that protect CIA development by controlling Th1 function [24]. They also inhibit TNFR2, which protects against inflammation through Treg function, as described above [13,28,31,32,33,34]. Thus, TNF blocking carries a risk of inhibiting the activity of some suppressor cells, and we sometimes encounter exacerbation of autoimmune diseases such as psoriasis, lupus-like syndrome, multiple sclerosis, and sarcoidosis during anti-TNF treatment [53]. Reagents such as specific inhibitors of TNFR1 could overcome these problems [54]

## 3. IL-6 and RA

IL-6, another key regulator of RA, was originally identified in 1986 as a secreted factor that induced immunoglobulin production [55]. Although several cell types can produce IL-6, including monocytes, T-lymphocytes, fibroblasts, and endothelial cells, IL-6 is secreted from synovial fibroblasts and B cells in the RA synovium [15].

### 3.1. Coordinated Interaction of TNF, IL-17, and IL-6 in RA Pathogenesis

A variety of stimuli, such as Toll-like receptor (TLR) ligands, IL-1β, and TNF can induce IL-6 transcription [55]. Sustained production of IL-6 after TNF-α stimulation in synovial fibroblasts is one of the features of RA [56]. F759 mice, which carry the Y759F mutation in glycoprotein 130 (gp130) and lack the negative feedback loop mediated by suppressors of cytokine signaling (SOCS) 3, develop arthritis spontaneously [55]. Genetic experiments have shown that F759 mice develop arthritis only if mutated gp130 functions in non-immune cells. Detailed experiments have shown that accumulation of IL-6 secreted from synovial fibroblasts leads to proliferation of CD4+ T cells, differentiation of Th17 cells, and subsequent development of arthritis [57]. This suggests that the pathogenesis of RA requires coordinated interaction of TNF, IL-17, and IL-6. In fibroblasts, TNF induces various types of cytokines and chemokines, whereas IL-17A alone does not induce either to any significant degree. Slowikowski et al. reported that TNF induced the expression of 370 genes, but that the expression of these genes was unaffected by addition of IL-17A. They also identified 26 genes whose expression was induced only upon co-stimulation with TNF and IL17-A, and 25 genes whose expression was induced by TNF and dose-dependently amplified by IL-17A; these included CXCL1, CXCL2, CXCL3, IL-6, IL-8, and MMP-3. This synergistic control is regulated by atypical inhibitor of kappa B (IκB) factor IκBζ, which is induced in proportion to IL17A concentration and functions dose-dependently [58]. Usually NF-κB binds IκB, and localizes to the cytosol, being referred to as classical or cytoplasmic IκB. When activated, classical IκB is degraded, leading to detachment of NF-κB from IκB and localization to the nucleus, thereafter, rapidly activating downstream genes. Upon activation of NF-κB, cytosolic IκB is degraded rapidly, but transcription of IκB is upregulated, and this re-synthesized IκB then inhibits cytosolic NF-κB, reverting to its basal state [59]. On the other hand, atypical IκB forms such as IκBζ localize to the nucleus upon stimulation and contribute to later transcriptional regulation after classical NF-κB [60]. IκBζ interacts with both NF-κB p65 and p100/52 in inflamed fibroblasts and coordinates the synergistic response to TNF and IL-17A. Synergistic induction of downstream genes such as IL-6 is canceled when IκBζ is repressed by siRNA [52].

### 3.2. Receptor–Ligand Interaction

IL-6 activates a signal cascade via three modes of receptor–ligand interaction, the first of which is classical signaling whereby IL-6 binds to its receptor, IL-6Rα, which is expressed on cells of lymphoid or myeloid lineage, thus activating an intracellular signal transduction pathway via gp130 dimerization. The second mode is trans-signaling. IL-6R also exists as a secreted form (sIL-6Rα), unlike the membrane-bound form, IL-6Rα, which is cleaved by proteases such as TACE. sIL-6Rα binds to IL-6, and the resulting complex binds to gp130 on endothelial cells and synovial fibroblasts, which usually do not express IL-6R. In the third mode, so-called trans presentation, circulating IL-6 binds to IL-6Rα expressed on dendritic cells (DC), and then the IL-6/IL-6Rα complex binds to gp130 expressed on CD4+ T cells. This mode of interaction is required for priming of Th17 cells [61]. Interaction of IL-6 and IL-6R and the signaling cascade are depicted in Figure 3.

### 3.3. Multidirectional Function of IL-6 in RA Pathogenesis

IL-6 is involved in a wide range of physiological processes, such as the immune response, inflammation, and bone metabolism, and has also been implicated in the pathogenesis of RA [1,17,55]. Here, we describe recent advances in knowledge regarding the regulation of Treg and Tfh cells, osteoclastogenesis, and VEGF production.

IL-6 regulates the differentiation of Treg, Th17, and Tfh cells. The IL-6-STAT3 pathway is required for Th17 cell development, and enhancement of the IL-6-STAT3 signaling axis causes IL-17A-dependent autoimmune arthritis in mice [62]. The importance of STAT3 in the differentiation of Th17 cells has also been demonstrated in humans [63]. Furthermore, IL-6 downregulates Foxp3 expression through STAT3 and induces the genetic reprogramming of Treg cells to Th17-like cells [64]. Thus, IL-6 regulates the Treg vs. Th17 cell balance. Conversion of Treg cells to Th17 cells is induced by IL-6 derived from synovial fibroblasts, and the converted Th17 cells are more osteoclastogenic than conventional Th17 cells [65].

T follicular helper (Tfh) cells have been identified as CXCR5+ PD-1+ CD4+ T cells, which regulate the differentiation of B cells into plasma cells and memory B cells by mediating class switching and affinity maturation of antibodies in germinal centers [66]. In mice, Bcl6+CXCR5+ Tfh differentiation is severely impaired in the absence of IL-6, suggesting that IL-6 is an essential factor for the development of murine Tfh cells [67]. In humans, however, Tfh differentiation might be less affected by IL-6, and TGF-β together with IL-12 or IL-23 induces various Tfh markers on CD4+ T cells [68].

IL-6 also regulates osteoclastogenesis in combination with TNF. Extensive analysis of osteoclast differentiation has identified various osteoclast types. Osteoclasts regulate bone absorption and bone mineral density (BMD), and cause bone erosion in both RA patients and model mice [22]. Differentiation of osteoclasts depends on TNF and RANKL, IL-6, and IL-17.

IL-6 also upregulates the expression of VEGF, which is also a pivotal cytokine in RA development [69]. Joint inflammation in RA is associated with angiogenesis, and migration of immune cells into the joint contributes to the pathogenesis of RA [70]. Although early administration of anti-VEGF antibody was reported to ameliorate vascularization and joint swelling in a CIA model mouse, later administration was not effective. These results suggest that angiogenesis via VEGF contributes to the early stage of RA pathogenesis. The synergistic effect of IL-6, IL-1β, and TNF on VEGF production is only impaired by anti-IL-6R antibody, but not by blockade of TNF or IL-1β, indicating that IL-6 is one of the major players in VEGF induction [71].

### 3.4. Evaluation of Anti-IL-6 Agents

IL-6 is involved in a wide range of physiological processes, such as the immune response, inflammation, and bone metabolism, and has also been implicated in the pathogenesis of RA. Anti-IL-6 receptor antibody was first approved for RA in 2008, and high efficacy in the treatment of RA has been demonstrated [48,49,50,51,52,55]. Due to the wide range of concentration of IL-6 in RA patients, clinical application of anti-IL-6 antibody seems to be difficult. In fact, clinical trials of the anti-IL-6 antibody sirukumab for RA were not successful [72]. The Food and Drug Administration (FDA) have declined the approval of this agent because of a trend for increased overall mortality with sirukumab vs. placebo. The mortality was mainly associated with cardiovascular events, infection, and malignancy. IL-6 activates the JAK-STAT system, and JAK inhibitors have a remarkable effect on RA [1,17,48,49,50,51,52]. Various cytokines regulate a number of downstream signaling molecules in RA, and these are potential therapeutic targets [10]. However, inhibition of pathways other than the JAK-STAT system is not considered to be easy due to the problem of crosstalk in which signals enter from other pathways.

### 3.5. RA-Related Comorbidities and Cytokine-Targeted Therapies

Cytokine-targeted therapies are also effective for RA-related comorbidities. Osteoporosis is one of the most common comorbidities of RA, and its prevalence is reported to be increased two-fold in RA patients [73,74]. As IL-6 and TNF contribute to osteoclastogenesis, blockade of IL-6 and TNF can protect against osteoporosis in RA patients. Some data support the idea that TNF inhibitors can improve or maintain BMD in RA patients [75,76,77,78]. Some reports have indicated that IL-6R blockade can have a favorable effect on BMD [79,80]. Furthermore, subgroup analyses of randomized clinical trials have indicated that sarilumab significantly decreases total RANKL levels, and that the ratio of RANKL to osteoprotegerin (OPG) is also significantly decreased in patients receiving sarilumab vs. adalimumab [81]. Chronic inflammation also influences cardiovascular disease [82]. Various studies have indicated that treatment with bDMARDs may be associated with a reduced risk of cardiovascular events [82,83,84]. Moreover, proinflammatory cytokines, especially IL-6, induce serum amyloid A, and can cause life-threatening secondary amyloidosis. Thus, bDMARDs treatment diminishes amyloidosis and dramatically improves the prognosis of affected patients [2].

## 4. IL-23/IL-17 and RANKL Elicit Bone Resorption by Driving the Function of Osteoclasts

### 4.1. IL-23 and IL-17

IL-23 is a member of the IL-12 cytokine family composed of the IL-23 p19 subunit and the IL-12/23 p40 subunit. It is secreted by activated macrophages and dendritic cells in peripheral tissues such as skin, intestinal mucosa, joints, and lungs [85,86].

IL-23 mainly induces the differentiation of αβ T CD4+ naïve cells (Th0 cells) into Th17 cells [85] and stimulates the production of proinflammatory cytokines such as TNF-α, IL-1β, IL-21, and IL-17 from Th17 cells, and IL-6 from macrophages and dendritic cells [87]. γδT cells and innate lymphoid cells constitutively express the IL-23 receptor (IL-23R). IL-23R is a heterodimeric receptor composed of two subunits: IL-12Rβ1 and IL-23Rα. The latter is specific to IL-23 signaling [85]. IL-23Rα interacts with JAK2, including STAT3 phosphorylation and leads to upregulation of retinoid-related orphan receptor gamma tau (RORγτ), facilitating the development of Th17 cells [88]. Th17 cells are present in synovial joints and secrete IL-17, which is a proinflammatory cytokine contributing to osteoclastogenesis along with TNF and IL-6 [89,90]. Recently, it has been reported that the proportion of receptor CCR6+ Th17 cells is increased in the peripheral blood of treatment-naïve patients with early RA [91]. In addition, higher frequencies of Th17 cells have been observed in the synovium of RA patients relative to OA patients [92].

IL-17 is involved in both early and established RA disease. It promotes activation of fibroblast-like synoviocytes (FLS), osteoclastogenesis, recruitment and activation of neutrophils, macrophages and B cells [93]. IL-17A is the first described member of IL-17 cytokine family, which includes six members, IL-17A to IL-17F [94]. TGF-β, IL-6, and IL-21 activate T lymphocytes and promote the initial differentiation of Th0 into Th17 cells, rendering them responsive to IL-23 [85]. These cellular events are crucial for Th17 cell stabilization and expansion.

The IL-17 receptor (IL-17R) is expressed on various cell types such as epithelial cells, B and T cells, fibroblasts, monocytic cells, and bone marrow stroma [85,93,94]. The main roles of IL-17 include the promotion and initiation of chemotaxis and the recruitment and activation of neutrophils in inflamed tissues. In inflammatory conditions such as inflammatory bowel diseases and arthritis, the serum and tissue levels of IL-17 are increased relative to non-pathological settings where IL-17A levels are extremely low or undetectable [85]. Synergism between IL-17 and TNF-α has been shown to activate the production of proinflammatory mediators such as IL-1β, IL-6, IL-8, prostaglandin E2, and MMPs, thereafter promoting progression of early inflammation toward chronic arthritis [85].

### 4.2. RANKL Regulation by Cytokines

The differentiation of osteoclasts is induced significantly in the presence of IL-17, either directly or indirectly through upregulation of RANKL [85].

Bone erosion by osteoclasts is one of the most important pathologic features of RA [95,96]. RANKL is a cytokine belonging to the TNF superfamily and stimulates mainly osteoclast differentiation. It binds to RANKL on osteoclasts and osteoclast precursors and promotes osteoclast differentiation and activation. OPG is a soluble decoy receptor of RANKL, and upon binding to RANKL inhibits its activation. The inflamed synovium and pannus in RA produce significantly higher levels of RANKL and lower levels of OPG in comparison to healthy synovium [97,98]. The cells responsible for the increased expression of RANKL in the inflamed synovial membrane are FLS and T lymphocytes. An increased RANKL/OPG ratio promotes osteoclast differentiation and activation at the synovium–bone interface and the development of bone erosions in RA [97,98].

RANKL on T cells or fibroblasts, which are activated by a combination of IL-6, IL-17, and TNF, regulates the differentiation of osteoclasts. IL-6 signaling induces RANKL ex-pression in RA-FLS through the expression of nuclear factor of activated T cells (NFATc)1 and tartrate-resistant acid phosphatase (TRAP) 5b mRNA in co-cultures of RA-FLS and osteoclast precursor cells [99].

IL-17 increases the expression of RANKL in adjuvant-induced arthritis-derived synovial fibroblasts, leading to increased osteoclastogenesis in vitro. As both RANKL expression and osteoclastogenesis are reduced by blocking IL-17R and STAT3, they are dependent on these molecules [100]. Furthermore, IL-17 modulates osteoclast precursor cells. Raw264.7 cell culture experiments have shown that a low level of IL-17A promotes the RANKL-RANK system by mediating the c-Jun N-terminal kinase (JNK) signaling pathway and activating autophagy and osteoclastogenesis in induced osteoclast precursor cells. However, a high level of IL-17A inhibits osteoclastogenesis [101]. A RANKL-independent osteoclast differentiation pathway has also been reported.

TNF induces differentiation of osteoclasts from mouse bone marrow myeloid cells (mBMM) and human peripheral blood monocytes (PBMC) independently of RANKL function [102]. TRAF3 limits RANKL-induced osteoclast formation by promoting proteasomal degradation of NF-κB-inducing kinase in a complex with TRAF2 and cellular inhibitor of apoptosis protein (cIAP). It also inhibits osteoclast formation induced by TNF. Hydroxychloroquine is an anti-inflammatory drug for RA that prevents TRAF3 degradation in osteoclast precursors and inhibits osteoclast formation in vitro [103].

TRAP-positive cells induced by TNF lack bone absorption ability but gain it upon treatment with a combination of IL-6 and TNF. Thus IL-6 is required for the bone-resorbing activity of TRAP-positive cells induced by TNF. Interestingly, whereas bone absorption activity cannot be abolished in mice with STAT3 conditional knockout, treatment with JAK inhibitor or MEK inhibitor achieves this [104]. Furthermore, the number of osteoclasts induced ex vivo from RA patient PBMCs with TNF and IL-6 is positively correlated with the host patient modified total Sharp score [105]. IL-17 A also promotes MMP production in chondrocytes [106].

### 4.3. Evaluation of Anti-IL-17 Agents

Antibodies against IL-17 (ixekizumab and secukinumab) or IL-17R (brodalumab) have been examined in patients with RA [107,108,109,110,111]. In a phase I randomized controlled trial (RCT) of RA patients treated with oral DMARDs, the addition of ixekizumab improved RA signs and symptoms and disease activity score (DAS) 28, compared to placebo [107]. This improvement was confirmed in a phase II study in which ixekizumab was administered to patients who were naïve to biological therapy or resistant to TNF-α inhibitors [111]. In a phase II study enrolling RA patients with an inadequate response to methotrexate, greater decreases in DAS28 were observed with secukinumab than with placebo [108].

There are a few reports based on meta-analyses of anti-IL-17 antibody agents. Kunwar et al. conducted a systematic review of studies retrieved by a sensitive search strategy in PubMed, EMBASE and Cochrane CENTRAL from inception through 9/7/15 [112].

Seven studies involving a total of 1226 patients, including 905 in an anti-IL-17 group and 321 in a placebo group, were included in the meta-analysis. Anti-IL-17 was effective in achieving ACR20 and ACR50 relative to placebo (odds ratio (OR) 2.47, 95% CI 1.29–4.72, *p* = 0.006, I2 77% and OR 2.94, 95% CI 1.37–6.28, *p* = 0.005, I2 64%, respectively). Data analysis for ACR70 showed a favorable trend toward anti-IL-17 (OR 2.62, 95% CI 1–6.89, *p* = 0.05, I2 15%). Subgroup analysis of ACR20 for individual anti-IL-17 agents showed that ixekizumab was more effective than placebo, while secukinumab showed a trend toward achieving the ACR20 response. However, brodalumab was not effective relative to placebo. Safety analysis showed no increased risk of any or serious adverse effects of anti-IL-17 compared to placebo (OR 1.23, 95% CI 0.94–1.61, *p* = 0.13, I2 = 0% and OR 1.28, 95% CI 0.57–2.88, *p* = 0.55, I2 = 0%, respectively). It was concluded that anti-IL-17 is effective for the treatment of RA without any increased risk of serious adverse effects; however, the results were limited by the significant heterogeneity and short duration of the studies [112].

In a recent meta-analysis, Wu et al. divided 2499 RA patients into two subgroups: those that were biologic-naïve and those that showed an inadequate response to TNF inhibitor. For the biologic-naïve patients, ACR50 and ACR70 responses were significantly better with IL-17 inhibitors than with placebo (RR = 1.71, 95% CI 1.232313–2.38, *p* = 0.001 and RR = 2.63, 95% CI 1.10–6.25, *p* = 0.03, respectively), but ACR20 responses for IL-17 inhibitors were not significantly superior to those for placebo (RR = 1.34, 95% CI 0.94–1.91, *p* = 0.11). For TNF-IR, IL-17 inhibitors were effective in achieving ACR20 (RR = 1.67, 95% CI 1.40–2.00, *p* < 0.), ACR50 (RR = 1.94, 95% CI 1.43-2.63, *p* < 0.0001) and ACR70 (RR = 2.11, 95% CI 1.26–3.55, *p* = 0.005) compared to placebo. Safety analysis showed that IL-17 inhibitors had no increased risk of any AEs relative to placebo in both biologic-naïve patients and TNF-IR patients. It was concluded that IL-17 inhibitors were effective for treatment of RA without any increased risk of AEs, in either biologic-naïve patients or TNF-IR patients [113].

Huang et al. performed a meta-analysis of the efficacy and safety of secukinumab in comparison with placebo in patients with active RA showing an inadequate response to TNF inhibitors. A total of 1292 patients from three phase III RCT studies were examined. The group receiving an injection of secukinumab (150 mg) showed a superior outcome in terms of ACR 20 risk ratio (1.66, 95% CI 1.33, 2.08; *p* < 0.0001; I2 = 0%), ACR50 (1.88, 95% CI 1.29, 2.72; *p* = 0.0009; I2 = 0%), and ACR70 (2.15, 95% CI 1.15, 4.02; *p* = 0.02; I2 = 0%) at 24 weeks compared with placebo. In the pooled secukinumab group, neither an increased risk nor any serious adverse events were detected at 16 weeks. They concluded that secukinumab may be a therapeutic option for patients with active RA who show an inadequate response to TNF inhibitors [114].

Lastly, double blockade of IL-17A and IL-17F (bimekizumab) in RA patients with an inadequate TNF-α response has been shown to achieve a greater reduction of DAS28-CRP at week 20 relative to comparable patients receiving a placebo [115].

### 4.4. Clinical Evaluation of Anti-IL-23 Agents

IL-23 may be a biomarker of human RA [116]. However, treatment with IL-23 inhibitor did not significantly reduce the signs and symptoms of RA.

The efficacy and safety of guselkumab (a human IgG1 antibody against the p19 subunit of IL-23) and ustekinumab (a human monoclonal anti-IL-12/23 p40 antibody) were evaluated in patients with active RA who were unresponsive to methotrexate therapy. At week 28, no significant difference in the proportions of patients who achieved an ACR 20 response was evident between the combined ustekinumab group (53.6%) or the combined guselkumab group (41.3%) relative to placebo (40.0%) (*p* = 0.101 and *p* = 0.877, respectively). Treatment with ustekinumab or guselkumab did not significantly reduce the signs and symptoms of RA. No new safety findings were observed with either treatment [117].

### 4.5. IL-7 and IL-21

IL-7 is expressed by stromal cells in primary lymphoid organs and is known to play a critical role in the development and homeostatic expansion of T cells in humans and mice [118]. IL-7 is overexpressed in inflamed tissues of patients with rheumatic autoimmune diseases and the level of IL-7 expression is associated with clinical parameters of the disease [119]. The differentiation of Th17 cells is mediated mainly by STAT3 signaling through cytokines such as IL-6, IL-21 and IL-23. In the phase of T cell activation and differentiation, the IL-7 receptor (IL-7R) is expressed on Th17 cells. In the differentiation phase, IL-7R is re-expressed in activated Th17 cells and IL-7 is critically requisite for sustaining the survival and expansion of differentiated Th17 cells through STAT5 signaling [120].

IL-21 has multifaceted roles in activating not only T cells but also B cells, monocytes/macrophages and synovial fibroblasts in RA pathogenesis through activation of the JAK/STAT, MAPK, and PI3K/Akt signaling pathways [121,122]. ATR-107, a fully human monoclonal anti-IL-21 receptor (IL-21R) antibody, has been examined for its effects in healthy subjects [123]. It showed a prolonged pharmacodynamic effect measured in terms of IL-21R occupancy and was highly immunogenic after single-dose administration, showing rapid clearance and low bioavailability. On the other hand, NNC0114-0005, a human recombinant anti-IL-21 monoclonal antibody for treatment of RA, was assessed in 20 RA patients treated with methotrexate in comparison with 44 healthy subjects (HS). Single doses of NNC0114-0005 (≤25 mg/kg IV; ≤4 mg/kg SC) were well tolerated in both HS and patients with RA. Accumulation of IL-21-containing complexes has suggested neutralization of the target cytokine [124]. Although GSK2618960, an IL-7 receptor-α subunit (CD127) monoclonal antibody, has been tested in healthy subjects, it has never been applied for patients with RA so far [125].

## 5. bDMARDs, JAK/STAT Inhibitors and Potential Molecular Targets for Treatment of Bone Resorption in RA

### 5.1. bDMARDs and Bone Metabolism

A previous immunohistochemical study using synovial samples from 18 RA patients has evaluated the expression of OPG and RANKL protein. After 8 weeks of treatment, infliximab and etanercept were found to increase the expression of OPG in synovial tissue, but neither influenced the expression of RANKL. In groups of patients treated with TNF inhibitor, the RANKL:OPG ratio decreased following therapy, suggesting that TNF inhibitors in RA modulate the OPG/RANKL system, possibly explaining the retardation of radiographically evident damage [126].

In another study, BMD in the spine, hip, and hand, as well as serum RANKL and RANKL/OPG, were monitored for 12 months in 102 RA patients receiving infliximab. BMD in the spine and hip showed no significant change, but BMD in the hand showed a significant decrease. The serum RANKL level and RANKL/OPG ratio were significantly decreased by infliximab treatment [127]. On the other hand, infliximab directly promotes the differentiation of osteoclast precursor cells in PBMC and lacunar resorption induced by RANKL. Addition of infliximab has been shown to markedly increase the number of TRAP-positive multinucleated cells (TRAP+ MNCs) and the extent of lacunar resorption in comparison with control cultures [128,129].

IL-6 and soluble IL-6 receptor (sIL-6R) induce RANKL expression in RA-FLS. Although IL-17 and TNF-alpha stimulate cell growth and IL-6 production in RA-FLS, they do not induce RANKL expression. IL-6 and sIL-6R induce the expression of NFATc1 and TRAP5b mRNA in cocultured RA-FLS and osteoclast precursor cells [99].

In mouse calvarial osteoblasts, IL-6 and sIL-6R induce bone resorption, and this is decreased by osteoclast inhibitors. Thus, IL-6 signaling influences osteoclastogenesis [130]. Tocilizumab significantly ameliorates bone erosion in metacarpal bones of patients with RA when injected weekly for 52 weeks subcutaneously, whereas adalimumab and methotrexate do not ameliorate bone erosion to any significant degree [131]. Tocilizumab suppresses the number of histologically evident osteoclasts and the degree of RANKL-induced bone erosion in the metacarpal bones of model monkeys with CIA, indicating that IL-6/IL-6R is involved in subchondral bone and bone marrow change in RA patients [132].

Abatacept, a chimeric molecule comprising the extracellular domain of the co-inhibitory molecule CTLA-4 fused to the Fc portion of a human IgG1 antibody, neutralizes binding of the CTLA-4 part to either CD80 or CD86 on the surface of activated antigen-presenting cells [133,134]. Abatacept dose-dependently inhibits RANKL-mediated osteoclast formation in monocytes, exerting an anti-bone resorbing effect [135].

Recent studies have clarified the action of abatacept on osteoclastogenesis in more detail. Bozec et al. have described a mechanism for control of bone resorption by the adaptive immune system and showed that CD80/86 negatively regulates the generation of bone-resorbing osteoclasts. In CD80/86-deficient mice with osteopenia due to increased osteoclast differentiation, inhibition of CD80/86 by administration of CTLA-4 activated the enzyme indoleamine 2,3-dioxygenase (IDO) in osteoclast precursors, causing degradation of tryptophan and promotion of apoptosis [136]. CTLA4-Ig inhibits osteoclast differentiation and reduces the expression of NFATc1 in bone marrow macrophages. It also suppresses calcium oscillations dependently on FCgammaR [137].

### 5.2. JAK/STAT Inhibitors and Bone Metabolism

The JAK/STAT signaling pathway appears to be related to bone homeostasis. JAK protein associated with the receptor after formation of the ligand–receptor complex is activated by transphosphorylation. JAK activation induces the phosphorylation of tyrosine on a cytoplasmic tail subunit of the receptor at docking sites for STAT proteins. The STAT proteins then undergo phosphorylation and dimerization, and the dimer translocates into the nucleus, where it binds to DNA and activates the transcription of targeted genes that affect cell behavior. SOCS provides negative feedback to the receptor and prevents continuous signaling. JAK1 and STAT3 signaling is mediated by IL-6 family cytokines, which bind to the gp130 IL-6 receptor subunit and are indispensable for normal skeletal development in mice and humans [138]. IL-6 family cytokines such as IL-6, IL-11, oncostatin M (OSM), cardiotropin-1 (CT-1), and leukemia inhibitory factor (LIF) stimulate activation and production of osteoclast via osteoblast lineage cells [138]. They also stimulate bone formation given that LIF increases trabecular bone mass in vivo and that LIF, IL-6, OSM, and IL-11 act directly on osteoblasts in vitro [138]. G-CSF also stimulates bone formation [139]. The JAK1/STAT3/SOCS3 pathway is activated by these cytokines and promotes both osteoblast and osteoclast formation [139,140]. Tofacitinib (a JAK1/JAK3 inhibitor) dose-dependently reduces RANKL expression in cultured T cells [141]. In a rat adjuvant-induced arthritis model, tofacitinib has been reported to increase bone cortical and trabecular hardness, but it does not reverse the effects of arthritis on cortical and trabecular bone structure [142].

Stattic, a STAT3-specific inhibitor, suppresses STAT3 and NF-κB, resulting in inhibition of RANKL-mediated osteoclastogenesis in RANKL-induced Raw264.7 cells. Stattic also inhibits bone loss caused by ovariectomy [143]. Baricitinib (a JAK1/JAK2 inhibitor) exerts an inhibitory effect on osteoclastogenesis. It suppresses RANKL expression in murine calvaria-derived osteoblasts. Furthermore, shRNA-mediated knockdown of JAK1 or JAK2 suppresses RANKL expression in osteoblasts and inhibits osteoclastogenesis. Therefore, it has been suggested that JAK1 and JAK2 represent novel therapeutic targets for osteoporosis as well as RA [144].

### 5.3. Reactive Oxygen Species and Bone Metabolism

In RA patients, reactive oxygen species (ROS) are highly expressed in neutrophils and synovium [145]. RANKL itself induces nitric oxide synthase (NOS), and N-acetyl cysteine (NAC) inhibits RANKL-induced ROS production and differentiation of osteoclasts in bone marrow monocyte-macrophage lineage cells [146]. Osteoclasts are activated by ROS to drive bone resorption [147]. In RA synovial fibroblasts, NAC attenuates the expression of RANKL mRNA and production of soluble RANKL in an IL-17 dose-dependent manner. IL-17 enhances the phosphorylation of mTOR, JNK, and IκB-α [148]. NAC inhibits both ROS and MMP-3 mRNA by interfering with the JNK signaling pathway [149]. Thus, JNK may have potential as a target for intervention in RA patients.

## 6. GM-CSF and the Pathogenesis of RA

GM-CSF is a well-known hemopoietic growth factor produced mainly by T cells and stromal cells. It is also essential for regulating the functions of mature myeloid cells such as macrophages. As well as its stimulating effects on mature granulocytes, it induces the expression of HLA class II-antigen on synovial cells in patients with RA [150]. GM-CSF levels are increased in the serum, synovial fluid and bone marrow of patients with RA, especially at the chronic stage [151,152]. IL-2, IL-7, and IL-33 are known to regulate the function of type 2 innate lymphoid cells [153]. In addition, the production of IL-33 is significantly increased in inflamed joints relative to healthy controls, and IL-2 is supplied from activated Th17 cells [154]. Additionally, production is synergistically increased when IL-33 and IL-2 are applied in combination [154].

Myeloid cells, T and B cells, and tissue resident cells can secrete GM-CSF, which in turn is capable of polarizing the function of macrophages into M1-like production of inflammatory cytokines. These cytokines induce the recruitment of inflammatory cells and activation of tissue resident cells. GM-CSF induces the production of IL-6 and IL-23 by antigen-presenting cells. IL-6 and IL-23 cause activation of T cells and their differentiation to Th17 cells, which in turn secrete GM-CSF and IL-17, thus maintaining the cycle. GM-CSF produced by Th17 cells also induces inflammation by activating the monocyte-macrophage system and neutrophils [155]. Macrophage populations in synovial tissue are associated with articular damage [156]. GM-CSF receptor activation leads to downstream involvement of JAK-STAT-SOCS as well as other pathways involving mitogen-activated protein kinases (MAPK), PI3K, and NF-κB [157,158].

Clinical trials of agents targeting the GM-CSF pathway in RA have been reported. Mavrilimumab is a human monoclonal antibody that inhibits the human GM-CSF receptor [159]. A phase 2a trial of mavrilimumab at doses of up to 100 mg found that 55.7% of the subjects met the primary endpoint of a ≥1.2 decrease from baseline disease activity scores at week 12 (vs. only 34.7% of subjects given a placebo) [160].

Theoretically, the signal of GM-CSF could be controlled by blocking the signal transduction pathway, which involves JAK-STAT and MAPK. Jak2 inhibits the GM-CSF signaling pathway by acting on JAK-STAT. For the treatment of RA, peficitinib is an oral JAK inhibitor that targets JAK family enzymes (JAK1, JAK2, JAK3, TYK2) and blocks the signal transduction of various cytokines, including GM-CSF [161]. Baricitinib is another inhibitor exerting similar effects [162]. Since these two drugs also inhibit JAK1 and block the IL-6 signaling pathway, any inhibitory effect on GM-CSF alone remains unclear.

## 7. IL-1β and the Pathogenesis of RA

The IL-1 cytokine family comprises 11 members that promote the activity of cells of the innate immune system, such as neutrophils, eosinophils, basophils, mast cells and natural killer cells [163]. While most of these cytokines are biologically active as full-length molecules, activation and secretion of IL-1β and IL-18 requires inflammasome/caspase-1-dependent processing [164]. In this section, we focus on IL-1β, IL-18, and IL-33, which are strongly associated with RA. IL-1β is considered to serve a crucial role in the progression of RA, being mainly produced by monocytes and macrophage cells. Natural killer (NK) cells, T cells, B cells, endothelial cells, synovial cells, and neutrophils also produce IL-1β. IL-1β activates monocytes/macrophages, thus leading to increased inflammation. It also induces proliferation of fibroblasts, and thus the synovium. Additionally, IL-1β activates chondrocytes, leading to cartilage damage, and activates osteoclasts, leading to bone resorption [165]. Natural IL-1 receptor antagonist (IL-1Ra) is able to block the binding of a cytokine from the same family to its receptor. IL-1Ra, produced from monocytes and macrophages together with IL-1β, suppresses IL-1 activity by competitively inhibiting the binding of IL-1 to the cell surface receptor, thus preventing IL-1β from sending a signal to the cell [166]. In RA patients, the plasma and synovial fluid concentrations of IL-1 are elevated, and this correlates with various parameters of disease activity. The production of endogenous IL-1Ra appears to be insufficient to balance these higher IL-1β levels [167]. There are four possible mechanisms of IL-1 inhibition. The first is inhibition by anti-IL-1 antibody, the second is IL-1 receptor inhibition, the third is inhibition of IL-1 synthesis, and the fourth is IL-1 receptor signal inhibition. Since intracellular signal transduction involving the IL-1 cytokine family is mediated by several kinases, crosstalk cannot be suppressed and it is difficult to block signal transduction with JAK inhibitors. Anti-IL-1 antibody and IL-1Ra can be considered for suppression of IL-1 under such conditions. The efficacy of IL-1β blocking in patients with active RA has been established in controlled clinical trials of anakinra, a recombinant human IL-1Ra (r-metHuIL-1ra). When used alone or in combination with methotrexate, anakinra significantly reduces the clinical signs and symptoms of RA compared with placebo. Taken together, these results indicate that IL-1β plays an important role in the pathogenesis of RA [168]. In addition, the anti-IL-1β neutralizing monoclonal antibody canakinumab and the soluble decoy receptor rilonacept have been approved. These two therapeutics have been used to block the role of IL-1 in numerous diseases but were not originally approved for RA [169,170]. As these IL-1 blockers might have a weaker clinical effect than TNF-α inhibitors [171,172], no large trials for RA treatment have been planned.

## 8. IL-18 and the Pathogenesis of RA

IL-18 is a pleiotropic cytokine involved in regulation of the innate and acquired immune responses. IL-18 requires post-translational processing by the cytoplasmic enzyme caspase-1 to an 18 kDa biologically active mature form. IL-18 binds to its specific receptor IL-18 alpha chain (IL-18Rα), forming a low-affinity ligand chain [173]. IL-18 plays a prominent role in the onset and maintenance of an inflammatory response during RA. It has been reported that IL-18 expression in synovial tissue correlates with the severity of joint inflammation and the levels of TNF-α and IL-1β. It is also considered that expression of IL-18 in rheumatoid synovial tissue correlates with the acute phase response [174]. IL-18, IL-1β, or TNF-α can indirectly stimulate osteoclast formation through upregulation of RANKL production from T cells in RA synovitis; IL-18 is as effective as IL-1β, but less potent than TNF-α [175]. IL-18 is predominantly associated with enhancement of IFN production in IL-12- or IL-15-related mechanisms [176]. Due to the strong proinflammatory activity of IL-18, many researchers are investigating its therapeutic potential in inflammatory diseases. To neutralize IL-18, interleukin-18-binding protein (IL-18 BP) or anti-IL-18 Ab formulations have been devised and clinical trials have been conducted to verify their safety and efficacy [177,178]. Currently, IL-18 BP shows promise for use in adult-onset Still’s disease and NLR Family CARD Domain Containing 4-related macrophage activation syndrome. For intracellular signaling, the IL-18 receptor (IL-18Rα) forms a heterodimer with IL-18Rβ to mediate signaling after binding to IL18 and downstream signaling to NF-κB and p38MAP [179]. Since IL-18 intracellular signal transduction is also mediated by several kinases, crosstalk cannot be suppressed and it is difficult to block signal transduction with JAK inhibitors. Recently, a study using a murine CIA model has suggested that inhibition of cytokine signaling by the SOCS family constitutes a major negative feedback mechanism to prevent runaway inflammation [180]. The transcription of SOCS proteins is rapidly upregulated in cells stimulated with cytokines. The SOCS proteins then reduce the impact of cytokines by interacting with JAKs [181]. On the other hand, nitric oxide inhibits caspase-1 activity, providing a potential feedback loop whereby IL-18 may regulate its own cleavage, although its indication for treatment of RA remains unclear [182].

## 9. IL-33 and the Pathogenesis of RA

IL-33, a member of the IL-1 family, is recognized as a ligand for the ST2 receptor [183]. IL-33 binds to a specific heterodimeric cell surface receptor, which is a dimer of IL-1Receptor associated protein (IL-1RacP) and ST2. ST2 exists in two forms as splice variants: a soluble form (sST2), which acts as a decoy receptor, sequesters free IL-33, and does not signal, and a membrane-bound form (ST2), which activates the NF-κB and MAPK signaling pathway to enhance the functions of mast cells, Tregs, and innate type 2 lymphoid cells [184]. The IL-33 protein is expressed mainly in epithelial and endothelial cells, particularly in high endothelial venules in human synovial tissue, and in cultured human RA fibroblasts [185]. IL-33 expression is strongly induced by IL-1 and/or TNF-α. Furthermore, IL-33 is highly expressed and its mRNA also detected in the joints of CIA mice, increasing during the early phase of the disease. In the CIA model, and in arthritic mouse joints, inhibition of IL-33 signaling attenuates the severity of experimental arthritis. Administration of a blocking anti-ST2 antibody at the onset of disease has been reported to attenuate the severity of CIA and reduce the degree of joint destruction. In addition, anti-ST2 antibody treatment is associated with a marked decrease in interferon production as well as with a more limited reduction of IL-17 production by ex vivo-stimulated draining LN cells, and levels of RANKL mRNA in the joint are reduced [185]. Despite promising findings from animal models, no clinical trials have yet been conducted in humans. In RA patients, IL-33 levels are increased in serum and synovial fluid [186,187] and associated with disease activity [188] and bone erosion [189]. One study has demonstrated that anti-IL-33 treatment also significantly decreased the serum levels of IFN-γ, IL-6, IL-12, IL-33, and TNF-α [190]. It is widely known that suppression of IL-6 and TNF-α directly prevents joint destruction. The network of the IL-33 signaling pathway was recently revealed, and the findings suggest that it may affect a wide variety of diseases, although it is complex [191]. The activity of IL-33 is mediated by binding to the IL-33 receptor complex (IL-33R) and activation of NF-κB signaling via the classical myeloid differentiation factor 88(MyD88)/IL-1 receptor associated kinase (IRAK)/TRAF6 module. IL-33 also induces phosphorylation and activation of the extracellular signal regulated kinase (ERK)1/2, JNK, p38 and PI3K/AKT signaling modules, resulting in the production and release of proinflammatory cytokines [191]. It has been shown that IL-33 induces memory-type pathogenic Th2 (Tpath2). Tpath2 induces chronic inflammation through the subsequent process in chronic pulmonary disease [192]. Theoretically, the IL-33 signal should be controllable by blocking the transduction pathway. In reality, however, inhibition of pathways other than the JAK-STAT system is not considered to be easy due to the problem of crosstalk in which signals enter from other pathways. Nevertheless, intervention with anti-IL-33 antibody and T2/IL-33 signaling might be an effective therapeutic option for RA.

## 10. IL-2 and RA

IL-2 is a Th1 lymphocyte-derived cytokine, and the principal autocrine growth factor that promotes T cell activation and proliferation [193].

Clinical studies have shown that the serum IL-2 level is correlated with disease activity in RA [194,195]. We have also shown that the levels of IL-2 in serum and bone marrow are correlated with disease activity in mutilans-type RA [152]. The effects of IL-2 are mediated by cell surface receptors (IL-2 R) expressed on activated T cells. The serum sIL-2 R level in RA reflects disease activation, and a rising level may also predict exacerbation of disease activity [196]. On the other hand, most studies of cytokines in RA have failed to detect IL-2 protein in RA synovial fluid [197]. In the CIA model, IL-2 was reported to exert two opposite effects, namely a direct stimulatory effect and an indirect suppressive effect. It was demonstrated that administration of recombinant human IL-2 (rhIL-2) at or just before disease onset exacerbated the disease (days 21–28 after the first immunization), whereas rhIL-2 administered before disease onset (days 14–21 after the first immunization) inhibited the CIA. It was concluded that the indirect suppressive effect was mediated by IFN-γ because in mice treated with an anti-IFN-γ Ab, both early and late IL-2 administration exacerbated the CIA [198]. In addition, IL-2/anti-IL-2 monoclonal antibody immune complexes have been reported to inhibit murine CIA. As for IFN-γ, according to a recent study, CD8+ T cells are the dominant source of IFN-γ, which activates CD4^+^ T cells, synovial fibroblasts, and monocytes/macrophages. Monocytes/macrophages produce INF-γ, which enhances osteoclastogenesis leading to joint damage in RA [12,198]

Histopathological examination of joints has revealed inhibition of synovial cell proliferation and lower levels of IL-17, IL-6, and TNF-α [11]. Although these data may reflect the usage of different types of IL-2, they may also reflect a change in the cytokine cascade in the mouse CIA model between the onset stage and the later stage. IL-2, IL-7, and IL-33 are known to regulate the function of type 2 innate lymphoid cells [153]. Stimulation with IL-33 causes GM-CSF production, and stimulation with a combination of IL-2 and IL-33 increases it. GM-CSF activates monocyte-macrophages and leads to joint damage. In RA patients, serum IL-2 levels are not only correlated with disease activity and autoantibody levels, but also impact their Th17/Treg immune imbalance. Additionally, in patients with active RA, levels of NK cells are abnormally elevated, possibly due to high serum levels of IL-2 [195]. IL-2-stimulated NK cells, which produce GM-CSF, induce monocyte-macrophage activation leading to joint damage. As mentioned in the section on GM-CSF, the network in which IL-2 is produced from Th17 lymphocytes acts on NK cells and macrophages via GM-CSF. These networks characterize RA in the chronic phase. Recently, a single open clinical trial was conducted on the immunological and clinical benefits of low-dose IL-2 across 11 autoimmune disorders, including RA [199]. The dose of IL-2 and the treatment scheme employed selectively activated and expanded Tregs and was safe across different diseases and concomitant treatments.

## 11. Conclusions

Recent advances in molecular biology have greatly expanded our understanding of the pathophysiology of RA. We overviewed the cytokines and molecules involved, starting from RF and ACPA, which represent the cornerstone autoantibodies operating in RA, and examined the diverse roles of cytokines such as TNF-α, IL-6, IL-7, IL-17, IL-21, IL-23, IL-1β, IL-18, IL-33, GM-CSF, and IL-2.

TNF and IL-6 inhibitors have already produced remarkable clinical effects in many patients. GM-CSF inhibitors are also likely to be used as therapeutic agents. Recently, treatment of RA is shifting from biologic DMARDs that control extracellular cytokines to signal transduction inhibitors that control intracellular signaling. Further advances in treatment are expected as research on intracellular signal transduction progresses, and other cytokines may also have potential for future treatment avenues.

## Figures and Tables

**Figure 1 ijms-22-10922-f001:**
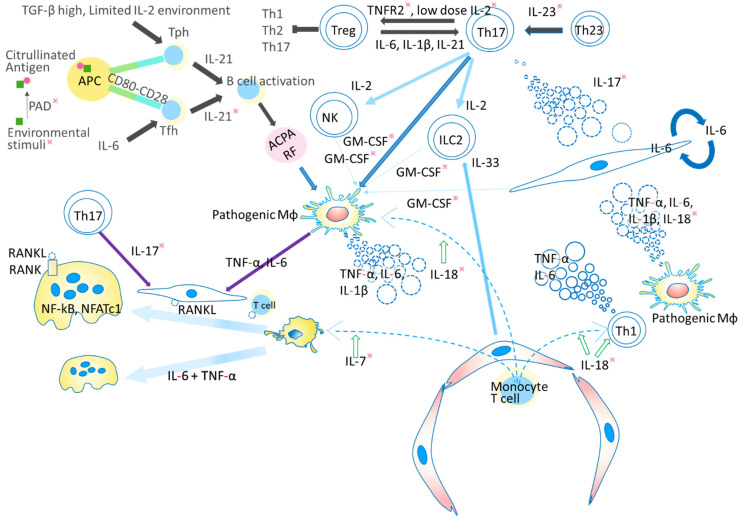
Possible mechanism of joint inflammation and destruction. In the preclinical stage (black arrows), environmental factors such as smoking, periodontitis and gut microbiota provide continuous antigen stimulation. Peptidylarginine deiminase (PAD) contributes to the production of citrullinated antigen and stimulates the production of autoantibodies such as rheumatoid factor (RF) and anti-citrullinated peptide antibody (ACPA). T follicular helper (Tfh) cells and T peripheral helper (Tph) cells influence the production of antigen by B cells. Interaction between T cells and antigen presenting cells (APC) through CD80-CD28 could be blocked by abatacept. In the early stage of RA development (dark blue arrows), ACPA forms an immune complex and activates macrophages to secrete proinflammatory cytokines such as tumor necrosis factor (TNF) and interleukin (IL)-6. In addition, self-antigen-stimulated Th17 cells secrete IL-17. Proinflammatory cytokines such as TNF, IL-6, and IL-17 activate synovial fibroblasts, which secrete various pathogenic molecules including IL-6, granulocyte macrophage colony-stimulating factor (GM-CSF), and matrix metalloproteinase (MMPs), leading to enrichment of proinflammatory cytokines. TNF and IL-17 induce RA pathogenic molecules synergistically. Furthermore, various signals such as IL-2 from T helper (Th) 17 cells, IL-7 from inflamed tissue, and IL-33 from endothelium or synovial fibroblasts can activate group 2 innate lymphoid cells (ILC2) to release GM-CSF. These GM-CSF act in the chronic phase (light blue arrows). These stimuli induce receptor activator of NF-κB ligand (RANKL) on fibroblasts and induce osteoclasts from extrinsic or intrinsic macrophages (purple arrows). RANKL binds to its receptor RANK and activates c-Jun through mitogen-activated protein kinase (MAPK) and c-Fos through NF-κB. Finally, it activates nuclear factor of activated T cell c1 (NFATc1) to induce the differentiation of osteoclasts, which play a role in bone resorption. Recent analysis has suggested that macrophages migrating from extra-synovial tissue play a pathogenetic role by secreting pro-inflammatory cytokines. On the other hand, tissue-intrinsic macrophages have an anti-arthritogenic function. IL-7 also potentiates osteoclast differentiation. All of the components illustrated here could potentially be targeted for future therapeutic applications, and some have already been put to practical use. Among them, promising candidates are indicated with an asterisk. Those with the most potential include escape from or control of environmental stimuli, inhibitors of cytokines other than TNF and IL-6, a specific stimulator of TNF receptor 2 (TNFR2), and a PAD inhibitor.

**Figure 2 ijms-22-10922-f002:**
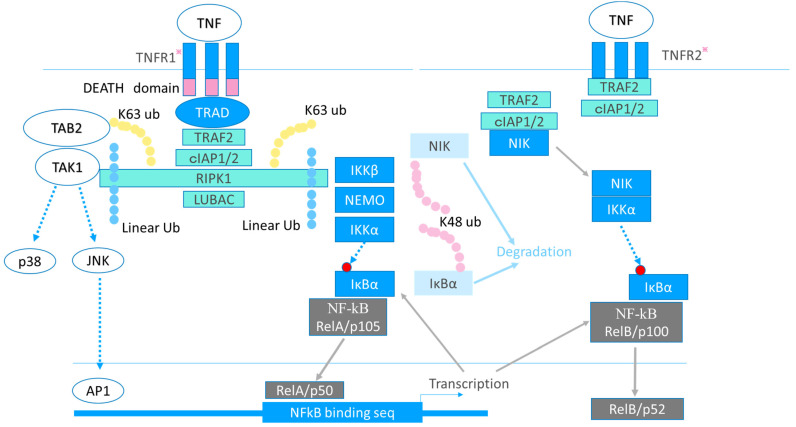
Signaling cascade of TNF. The tumor necrosis factor (TNF) receptor (TNFR)1 possesses the intracellular DEATH domain, which regulates apoptosis through activation of caspases, whereas TNRF2 does not. TNFR1 recruits the adaptor protein, TNF receptor-associated death domain (TRADD), through its DEATH domain. Activated TNFR1 recruits TRADD, TNF receptor-associated factor 2 (TRAF2), receptor-interacting protein kinase 1 (RIPK1), and cell inhibitor of apoptosis protein-1/2 (cIAP1/2) to form Complex I. Complex 1 activates NF-kB and triggers transcription of downstream inflammatory genes, including TNF itself, IL-6, IL-8, and IL-1β. Upon stimulation by TNF, a precisely controlled multistep ubiquitination process activates the downstream cascade. cIAP1/2 modify RIPK1 with K63-linked poly-ubiquitin (k63 ub). The linear ubiquitin chain assembly complex (LUBAC) is then recruited. LUBAC modifies RIPK1 with linearly linked ubiquitin chains (linear ub), then RIPK1 activates TGF-β-activated kinase 1 (TAK1) and the I kappa B kinase (IKK) complex that consists of IKKα, β, and the NF-κB essential modulator (NEMO). Usually NF-κB binds to the inhibitor of NF-κB α (IκBα) and is localized in the cytosol, because IκBα masks the nuclear localizing signal of NF-κB and interferes with NF-κB localization to the nucleus. The IKK phosphorylates IκBα, and then IκBα is K48-ubiquitinated, leading to its degradation. Finally, NF-κB becomes detached from IκBα and translocates to the nucleus. NF-κB activates the transcription of IκBα, and then newly translated IκBα binds to NF-κB to stop NF-κB activation. RIPK1 also activates c-Jun N-terminal kinase (JNK) and p38MAPK through TAK1. Although TNFR2 signaling also activates NF-κB and p38MAPK through a TRAF2-dependent mechanism as TNFR1, another signaling mechanism dependent on NF-κB inducing kinase (NIK) and independent of NEMO also functions. Usually, NIK is K48-ubiquitinated and constitutively degraded. Upon stimulation, NIK is released from cIAPs and activates IκBα, leading to the activation of NF-kB. In addition, consumeristic recruitment of TRAF2 by TNFR2 leads to depletion of cytosolic TRAF2 and perturbation of TNFR1-mediated signaling. Promising therapeutic targets are indicated with an asterisk. TNFR1 stimulation releases proinflammatory cytokines, and TNFR2 signaling has a protective function against arthritis. A specific inhibitor of TNFR1 or a specific activator of TNFR2 signaling, which could modulate the downstream signaling cascade, would be possible therapeutic targets. Promising candidates are indicated with an asterisk. TAB2: TAK1 binding protein1.

**Figure 3 ijms-22-10922-f003:**
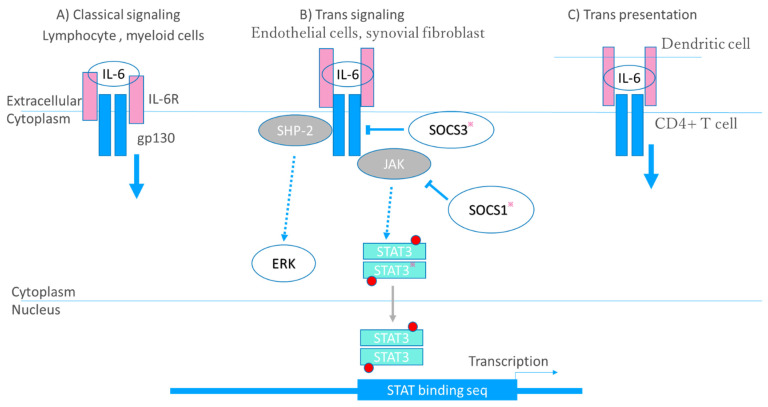
Signaling cascade of IL-6. Interleukin (IL)-6 activates a signal cascade via three modes of receptor–ligand interaction. (**A**) Classical signaling. IL-6 binds to its receptor, IL-6Rα, then activates an intracellular signal transduction pathway via glycoprotein130 (gp130) dimerization. (**B**) Trans-signaling. IL-6R also exists as a secreted form (sIL-6Rα). sIL-6Rα binds to IL-6, and the resulting complex binds to gp130 on endothelial cells or synovial fibroblasts. (**C**) Trans presentation. Circulating IL-6 binds to IL-6Rα expressed on dendritic cells, and then the IL-6/IL-6Rα complex binds to gp130 expressed on CD4^+^ T cells. The IL-6/IL-6Rα complex binds gp130 to form a gp130 homodimer, thus activating Janus kinase (JAK). JAK activates the signal transducer and activator of transcription (STAT) family transcription factors, mainly STAT3, and src homology region 2 domain-containing phosphatase 2 (SHP-2). JAK phosphorylates STAT3 to form a homodimer, which then translocates to the nucleus where it functions as a transcription factor. JAK also activates the extracellular signal-regulated kinase (ERK)/mitogen-activated protein kinase (MAPK) pathway through SHP-2. STAT3 induces suppressor of cytokine signaling1 (SOCS1) and SOCS3; SOCS1 inhibits JAK directly, and SOCS3 inhibits gp130. Promising therapeutic targets are indicated with an asterisk. In addition to a JAK inhibitor, an inhibitor of STAT, and a substrate that would regulate the level of SOCS expression could be therapeutic targets.

**Table 1 ijms-22-10922-t001:** Representative cytokines, signal transduction pathways, and their pathogenic roles in rheumatoid arthritis.

Cytokine-Receptor- Major Signaling Molecules	Proposed Roles in Rheumatoid Arthritis	Clinical Application
TNF-TNFR 1/2- NF-kB, MAPKs, PI3K	Osteoclastogenesis, TNFR1; Treg inhibition, Proinflammatory cytokine production TNFR2; Treg activation Epigenomic modification (acetylation/methylation) memTNF; protective against arthritis	Widely used
IL-6-IL-6R- gp130, JAK1/ 2, Tyk2, STAT1/3, PI3K, SHP-2, ERK	Osteoclastogenesis Proinflammatory cytokine production Autoantibody production Th17 differentiation, Treg inhibition	Widely used (anti-receptor antibody)
IL-33-ST2- IL-1-RacP, MyD88, IRAKs, TRAF6, NF-kB, MAPKs, AP1	Proinflammatory cytokine production Activation of mast cells, Tregs, Th2, and ILC2	Unreported
IL-1β- IL-1R, MyD88, IRAKs, TRAF6, NF-kB MAPKs, AP1	Inflammation Th17 differentiation, Treg inhibition	Modest or negative
IL-18-IL-18Rα/18Rβ MyD88, IRAKs, NF-kB IL-18–IL-18BP	Inflammation Neutralization	Unreported
IL-23–IL-12Rβ1/IL-23R- Tyk2, JAK2, STAT3/4	Activation of Th17, NKT, and ILC3 cells Cytokine production (IL-17, TNF-α, GM-CSF)	Did not meet primary endpoint
IL-17–IL-17R ACT1, TRAF6, NF-kB, MAPKs	Proinflammatory cytokine production Osteoclastogenesis Activation of synovial fibroblasts, macrophages	Did not meet primary endpoint
IL-7–IL-7R JAK1/3, STAT3/5, PI3K, AKT	Differentiation, expansion of Th17 cells Treg differentiation Osteoclastogenesis	Unreported
IL-21 JAK1/3, STAT1/3/5	Autocrine amplification of Th17 cells Th17 differentiation, Treg inhibition	Phase I/IIa
GM-CSF-GM-CSFR- JAK2, STAT3/5	Macrophage activation Proinflammatory cytokine production	Phase III
IL-2–IL-2R- JAK1/2/3, STAT3/5, SHC-1, ERK	Late phase: arthritogenic Activation of ILC2, NK cells, Th17 cells, IL-33 production Early phase: Anti-arthritogenic via IFN-γ Low dose IL-2 Treg activation	Phase I/IIa

Abbreviations: TNF, tumor necrosis factor; TNFR, tumor necrosis factor receptor; memTNF, membrane TNF; JAK, Janus kinase; Tyk2, tyrosine kinase 2; STAT, signal transducer and activator of transcription; PI3K, phosphatidylinositol-3 kinase; SHP-2, Src homology region 2 domain-containing phosphatase 2; ERK, extracellular signal regulated kinase; Th, T helper; Treg, T regulatory; IL, interleukin; IL-1-RacP, IL-1 receptor accessory protein; MyD88, *myeloid differentiation factor* 88; IRAKs, interleukin-1 receptor associated kinases; TRAF6, tumor necrosis factor receptor-associated factor 6; MAPK, mitogen-activated protein kinase; NF-κB, nuclear factor-kappa B; AP1, activator protein 1; ILC2, group 2 innate lymphoid cells; NKT, natural killer T cell; ACT1, nuclear factor activator 1; GM-CSF, granulocyte macrophage colony-stimulating factor; SHC-1, Src homology domain-containing transforming protein 1; IFN-γ, interferon-gamma.

## Data Availability

Not applicable.

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
