# Peer review of "Cytokine Networks in the Pathogenesis of Rheumatoid Arthritis"

_ijms, 2021, doi:10.3390/ijms222010922_

Round 1
Reviewer 1 Report
In this revised article, authors have addressed all comments. Given the conflict results regarding the effectiveness of IL-1β-targeting therapies, the use of “Effective” in Table 1 (Page 7) in IL-1 targeting therapies for RA patients may be inappropriate.
Author Response
Reviewer 1: Thank you for the excellent advice.
Given the conflict results regarding the effectiveness of IL-1β-targeting therapies, the use of “Effective” in Table 1 (Page 7) in IL-1 targeting therapies for RA patients may be inappropriate.
>>Thank you for your comment. As you indicated, efficacy of IL-1β-targeting therapies vary among drugs. Although, anakinra is approved for RA, canakinumab and rilonacept are not approved because of negative results. Furthermore, results of network meta-analysis indicated anakinra might be less effective than TNF-inhibitors and IL-6R antagonists (53, Janke et al. BMJ 2020). In this respect, we change “Effective” to “Modest or negative” in Table 1.
Reviewer 2 Report
In my opinion, the article may be published in this version.
Author Response
Reviewer 2: Thank you for the delightful comment.
We checked reference, and renumbering reference numbers in main text. We also performed English proofreading on your advice.
This manuscript is a resubmission of an earlier submission. The following is a list of the peer review reports and author responses from that submission.
Round 1
Reviewer 1 Report
In this review, authors provide an existing evidence regarding the pathogenic roles of cytokines, the related immune cells and the immune systemic in rheumatoid arthritis (RA), manipulation of which may offer the potential for more effective treatments in the future. The topic is relevant and interesting. Some comments are as the follows:
- As the authors’ description regarding the course of RA development, the very early preclinical stage to the established disease stage, it would be interesting to illustrate the involved cytokines or immune cells in the cytokine network in RA.
- It would be desirable if the authors could provide a meta-analysis of the results from post-marketed surveillance (PMS) or clinical practice for cytokine-targeting therapies based on the pathogenic cytokines in this review article.
- Given the crucial roles of cytokines in the pathogenesis of RA-related comorbidities such as RA-related osteoporosis as the described cytokines (TNF-α and IL-6) in this review article, it would be helpful if the authors could add the current status of the effectiveness of cytokine-targeted therapies on the RA-related comorbidities.
- It would be better to add the present and future targeted therapeutics in the Figure 1 (The possible mechanisms of joint inflammation and destruction) or add an additional figure to show the present and future targeted therapeutics in the cytokine networks in RA.
Author Response
As the authors’ description regarding the course of RA development, the very early preclinical stage to the established disease stage, it would be interesting to illustrate the involved cytokines or immune cells in the cytokine network in RA.
>> According to the review’s advice, we modified figure 1 and its description for very early preclinical stage to the established disease stage.
It would be desirable if the authors could provide a meta-analysis of the results from post-marketed surveillance (PMS) or clinical practice for cytokine-targeting therapies based on the pathogenic cytokines in this review article.
>> We added the results of PMS and clinical practice for cytokine-targeting therapies into the text.
Given the crucial roles of cytokines in the pathogenesis of RA-related comorbidities such as RA-related osteoporosis as the described cytokines (TNF-α and IL-6) in this review article, it would be helpful if the authors could add the current status of the effectiveness of cytokine-targeted therapies on the RA-related comorbidities.
>> According to the review’s advice, RA-related comorbidities into the text.
It would be better to add the present and future targeted therapeutics in the Figure 1 (The possible mechanisms of joint inflammation and destruction) or add an additional figure to show the present and future targeted therapeutics in the cytokine networks in RA.
>> According to the review’s advice, we added future targeted therapeutics in figure 1, 2, and 3.
Reviewer 2 Report
Kondo et al. have performed an interesting and valuable review of literature about some of the cytokines involved in the pathogenesis of rheumatoid arthritis, with an emphasis on the insightful description of a certain molecular pathways. Fragments according to cytokines themselves and signalling are good, however, they could be developed a bit further. Unfortunately presented article require rewritting, insightful intention for the readers, some major and minor corrections, and at this state is not suitable for publication.
Concerns:
- Some parts of the text seems a bit too convoluted and hard to read, thus require rewriting. This matter is especially visible in section 7. Examples:
- Line 511- “and is recognized as”.
- Lines 520-521- need to be rewritten.
- Lines 528-530- merge those 2 sentences.
- Line 538- It would be nice to mention some of those diseases.
- Line 574. Word “their” is redundant in that context.
- Line 394:”…following anti-TNF therapy”- redundant fragment.
- Line 441- It seems that after “and” some word has been lost.
- Lines 350-351, double usage of the word:” increased”, this could be changed.
- It would be easier to read if you could split some parts into separate sentences in ex. lines 110-113, 156- 161.
- Lines 359-360 :” As both are reduced by blockade of those molecules” would probably suit better.
- Section conclusions is a bit incoherent, It would be proper to rewrite it.
- Some sentences would require additional citations like in lines: 42-43, line 48 (lack of citation about DMARSs and JAK inhibitors in RA therapies), line 51 (lack of citation about the association of mentioned cytokines in chronic stages of RA), lines 254-255.
- In the graph (Figure 1.), some cytokines are written with a dash and some are not. This situation also occurs within the text (line 511) so you should decide on using one form.
- In the presented article authors describe cytokines involved in pathogenesis and exacerbation of RA, focusing on TNF-α, IL-6, IL-17, RANKL and GM-CSF. RA is a complex syndrome so perhaps it would be worth briefly mention and maybe develop topics other involved cytokines such as IL-18, IL-1, IL-7, IL-21.
- In the case of TNF-α and IL-6 authors presented detailed and valuable descriptions of protein characteristics, however in subsections about IL-2, IL-17 and GM-CSF those pieces of information are quite residual. Perhaps it would be worth developing those matters in case of other cytokines or to cut off the paper.
- In certain paragraphs, authors presented detailed and insightful descriptions of some cell’s signalling pathways. At the first glance, those matters may seem a bit convoluted so additional graphical presentation would be on plus.
- In Table 1. there is no need to develop some terms like “IL-33, interleukin 33” especially when u didn’t do that previously.
- IL-6 does also increase VEGF expression, which is also a pivotal cytokine in RA development: Guo, Q.; Wang, Y.; Xu, D.; Nossent, J.; Pavlos, N.J.; Xu, J. Rheumatoid arthritis: Pathological mechanisms and modern
pharmacologic therapies. Bone Res. 2018, 6, 1–14.
- It would be easier for the reader to keep up with information's about Iκβ if you could develop differences between functioning of “regular” Iκβ proteins and nuclear ones, with emphasis on their transcription inducing capabilities. Lines 221-224.
- In lines 249-251 it would be proper to already add information about fibroblast-like synoviocytes (FLS).
- Line 262- there is no information where the CD34+ population is located contrary to the rest.
- Lines 322-323. Those 2 sentences seem a bit inconsistent. From the context it looks like that those cells can migrate although they express different receptors to do so, this should be rewritten.
- In paragraph 3., you could also briefly mention different cells populations including for example chondrocytes.
- IL-17 does also affect chondrocytes, maybe it would be worth developing in section 4.
- Line 364- possibly via TRAF3 protein degradation.
- Line 378- The whole name of NOS protein would be required.
- Lines 381-382. This sentence is obscure, leaving an impression that IL-17 boost up NAC production, what according to the cited article is not true. The same matter occurs later in lines 465- 469. Part 465-469 should be merged with 381-382 and must be corrected.
- Lines 385-386, this sentence could be included in AtoM cells description – line 293.
- Section 5. – I would change the name of this paragraph into something like: “DMARDs, JAK/STAT inhibitors and potential molecular targets in RA therapy. The section about potential drugs should be presented as a last in the paper, before conclusions. Also, there is close to no information about osteoclastogenesis itself, so it would be better to move that part into RANKL or macrophages section.
- Line 447- the phrase “osteoclast activation” would suit better than “osteoclast formation”.
- Line 478- this line seems a bit out of context, try to merge it better within the text.
- Lines 498-508. Perhaps you should decide whether you want to describe all therapies in one section or to summarize each therapy with respective cytokines.
- Lines 562- 563. It would be nice to develop the relation between IL-2 and IFN-γ.
Some of the mentioned issues have been described in the paper recently published:
Makuch S, Więcek K, Woźniak M. The Immunomodulatory and Anti-Inflammatory Effect of Curcumin on Immune Cell Populations, Cytokines, and In Vivo Models of Rheumatoid Arthritis. Pharmaceuticals (Basel). 2021 Apr 1;14(4):309. doi: 10.3390/ph14040309. PMID: 33915757; PMCID: PMC8065689.
Therefore, to significantly contribute the field some fresh interesting insights should be proposed.
Author Response
According to the suggestion of reviewer, we will answer in order.
- Some parts of the text seems a bit too convoluted and hard to read, thus require rewriting. This matter is especially visible in section 7.
>> I'm sorry to bother you. We rewrote some parts according to the advice of the review.
- Some sentences would require additional citations like in lines: 42-43, line 48 (lack of citation about DMARSs and JAK inhibitors in RA therapies), line 51 (lack of citation about the association of mentioned cytokines in chronic stages of RA), lines 254-255.
>> We added citations according to the advice.
>>About lines 254-255, we deleted chapter 3 according to the comment of reviewer 2.
- In the graph (Figure 1.), some cytokines are written with a dash and some are not. This situation also occurs within the text (line 511) so you should decide on using one form.
>> Thank you for your comments. We unify the notation of cytokines with dash.
- In the presented article authors describe cytokines involved in pathogenesis and exacerbation of RA, focusing on TNF-α, IL-6, IL-17, RANKL and GM-CSF. RA is a complex syndrome so perhaps it would be worth briefly mention and maybe develop topics other involved cytokines such as IL-18, IL-1, IL-7, IL-21.
>> According to the review’s advice, we added citations according to the advice, we added IL-18, IL-1b, IL-7, IL-21 part into the text.
- In the case of TNF-α and IL-6 authors presented detailed and valuable descriptions of protein characteristics, however in subsections about IL-2, IL-17 and GM-CSF those pieces of information are quite residual. Perhaps it would be worth developing those matters in case of other cytokines or to cut off the paper.
>>According to your advice, we deleted the following sentences about TNF-alpha.
>>According to your advice, we deleted the following sentences about IL-6.
- In certain paragraphs, authors presented detailed and insightful descriptions of some cell’s signalling pathways. At the first glance, those matters may seem a bit convoluted so additional graphical presentation would be on plus.
>>Thank you for your advice. We added new figures about signaling pathways of TNF and IL-6. We added these sentences for the guide to the figures.
“Intracellular signaling process of TNF is depicted in Figure 2.” in line 93 of revised manuscripts
“Interaction of IL-6 and IL-6R and signaling cascade is depicted in Figure 3.” in line 244 of revised manuscripts
- In Table 1. there is no need to develop some terms like “IL-33, interleukin 33” especially when u didn’t do that previously.
>> According to the review’s advice, we corrected the text.
- IL-6 does also increase VEGF expression, which is also a pivotal cytokine in RA development: Guo, Q.; Wang, Y.; Xu, D.; Nossent, J.; Pavlos, N.J.; Xu, J. Rheumatoid arthritis: Pathological mechanisms and modern pharmacologic therapies. Bone Res. 2018, 6, 1–14.
>> Thank you for your adcive. We added the information about VEGF as follows in Line 269-.
“IL-6 does also increase VEGF expression, which is also a pivotal cytokine in RA development [64]. Joint inflammation of RA is associated with angiogenesis, and migration of immune cells into the joint contribute to the pathogenesis of RA [65]. Although early administration of anti-VEGF antibody ammeliorate vascularization and joint swelling in CIA model mouse, later administration did not improve arthritis. These results indicated angiogenesis via VEGF contribute early stage of RA pathogenesis. The synergistic effect of IL-6, IL-1β, and TNF on VEGF production is only impaired by anti-IL-6R antibody, but by blockade of TNF or IL-1β indicated IL-6 is one of the major player in VEGF induction [66].
- It would be easier for the reader to keep up with information's about IκB if you could develop differences between functioning of “regular” IκB proteins and nuclear ones, with emphasis on their transcription inducing capabilities. Lines 221-224.
>> Thank you for your advice. We added following sentences in Line 226 of revised manuscript -
“Usually NF-κB binds IκB, and localize in cytosol, and called classical or cytoplasmic IκB. When activated, these classical IκB is degradated. Then NF-κB detached from IκB and localized to nucleus, activating downstream genes in a rapid fassion. Upon activation of NF-κB, cytosolic IκB is degraded rapidly, but transcription of IκB is up-regulated, then this re-synthesized IκB inhibit cytosolic NF-κB, revert to its basal state [54]. On the contrary, atypical IκB such as IκBζ localize to the nucleus upon stimulation, and contribute to later transcriptional regulation after classical NF-kB[55]. They showed IκBζ interact both with NF-κB p65 and p100/52 in inflamed fibroblast, and coordinate synergistic response to TNF and IL-17A.
”
- In lines 249-251 it would be proper to already add information about fibroblast-like synoviocytes (FLS).
- Line 262- there is no information where the CD34+ population is located contrary to the rest.
- Lines 322-323. Those 2 sentences seem a bit inconsistent. From the context it looks like that those cells can migrate although they express different receptors to do so, this should be rewritten.
- In paragraph 3., you could also briefly mention different cells populations including for example chondrocytes.
>> Thank you for your comments. But we deleted chapter 3 following the suggestion of reviewer 2.
- IL-17 does also affect chondrocytes, maybe it would be worth developing in section
>> Thank you so much for the comment. We added the association of IL-17 in chondrocyte at Chapter 3 as follows in Line 381.
“IL-17 A also promotes MMP production in chondrocytes [93].”
- Line 364- possibly via TRAF3 protein degradation.
>> Thank you so much for the comment. TNF receptor associated factor-3 (TRAF3) limits RANKL-induced osteoclast formation by promoting proteosomal degradation. So, we added the information of TRAF3 by citing a new article as follows in Line 369;
“TNF receptor associated factor-3 (TRAF3) limits RANKL-induced osteoclast formation by promoting proteasomal degradation of NF-kB-inducing kinase in a complex with TRAF2 and cIAP. It also inhibits osteoclast formation induced by TNF. Hydroxychloroquine is anti-inflammatory drugs for rheumatoid arthritis and prevents TRAF3 degradation in osteoclast precursors and inhibits osteoclast formation in vitro [90].”
- Line 378- The whole name of NOS protein would be required.
>> Thank you for the comment. NOS means nitric oxide synthase. I added in the revised manuscript in Line 467.
- Lines 381-382. This sentence is obscure, leaving an impression that IL-17 boost up NAC production, what according to the cited article is not true. The same matter occurs later in lines 465- 469. Part 465-469 should be merged with 381-382 and must be corrected.
>> Thank you for the comment. As you pointed out, Line 381-382 and lines 465-469 are the same content. So, I deleted the former part related with oxidative stress and merged with the latter part in Chapter 4. The corrected sentence is as follows in Line from 466 to 475;
“In RA patients, ROS are highly expressed in neutrophils and synovium [118]. RANKL itself induces nitric oxide synthase (NOS), and N-acetyl cysteine (NAC) inhibits RANKL-induced ROS production and differentiation of osteoclasts in bone marrow monocyte-macrophage lineage cells [119]. Osteoclasts are activated by ROS to drive bone resorption [120]. In RA synovial fibroblasts, NAC attenuates the expression of RANKL mRNA and production of soluble RANKL in an IL-17 dose-dependent manner. IL-17 enhances the phosphorylation of mammalian target of rapamycin (mTOR), c-Jun N-terminal kinase (JNK), and inhibitor of kappaB alpha (IkB-α) [121]. NAC inhibits both ROS and MMP-3 mRNA by interfering with the JNK signaling pathway [122]. Thus, JNK may have potential as a target for intervention in RA patients.”
- Lines 385-386, this sentence could be included in AtoM cells description – line 293.
>> Thank you for your comment. I deleted Lines 385-386 not for repeating the content written in Line 293.
- Section 5. – I would change the name of this paragraph into something like: “DMARDs, JAK/STAT inhibitors and potential molecular targets in RA therapy. The section about potential drugs should be presented as a last in the paper, before conclusions. Also, there is close to no information about osteoclastogenesis itself, so it would be better to move that part into RANKL or macrophages section.
>> Thank you so much for the precious comment. We would like to mention about the influences of biological DMARDs and JAK/STAT inhibitors (targeted synthetic DMARDs) on bone remodeling (especially on bone resorption) in Chapter 5, so we corrected the subject as Chapter 4 in the revised manuscript into “DMARDs, JAK/STAT inhibitors and potential molecular targets for the intervention of bone resorption in RA in Line 392-393.”.
Line 447- the phrase “osteoclast activation” would suit better than “osteoclast formation”.
>> Thank you so much for the comment. We corrected “osteoclast activation” in the revised manuscript in Chapter 4 as follows in Line 452-453.
“The JAK1/STAT3/SOC3 pathway is also activated by G-CSF, thus inhibiting bone formation and promoting osteoclast activation [112,113].”
- Line 478- this line seems a bit out of context, try to merge it better within the text.
>> According to the review’s advice, we corrected the text.
- Lines 498-508. Perhaps you should decide whether you want to describe all therapies in one section or to summarize each therapy with respective cytokines.
>>Thank you for the advice. Since the control of cytokines was differs depending on the type, the treatment part was described separately.
- Lines 562- 563. It would be nice to develop the relation between IL-2 and IFN-γ.
>>Thank you for the advice. According to the review’s advice, we added the text.
Reviewer 3 Report
The pathogenesis of RA is still being explored, therefore the topic of the article is current.
Nevertheless the article is incoherent and requires significant amendments.
Most importantly, not all cytokines important in the pathogenesis of RA are included in the description. Proper presentation of the topic requires considering the full picture of the cytokine network.
Introduction
The introduction contains basic information about RA well-known and may be omitted.
Figure 1.
The figure is not clear. I propose to change it so that it really shows the cytokine network, which is the topic of the article.
Sections
The subsections are not clearly written. I suggest systematizing the descriptions, for example, by describing the source of cytokines and triggers, receptors (with localization), the effect of activation, and summarizing the effects on key joint cells. Then, add a reference to current and future therapies.
This information has been mixed, making it difficult to analyze the text.
Lines 70-104 describe the action of TNF receptors, but without reference to RA. I have not found any information on which cells are these receptors and how their activation relates to arthritis.
Line 107 “the cellular source of TNF and its relationship to the complex pathogenesis of RA has remained unclear”
I suggest looking for information on this topic.
Verses 135-169 describe epigenetic mechanisms. The entire section lacks information on how TNF is involved in the pathogenesis of subsequent stages of arthritis in RA (topic of article).
Section 3 describes the cellular structure of the joint synovium and does not address the subject of the manuscript. I suggest changing the section to relate to the topic.
Rheumatoid arthritis is mainly associated with chronic inflammation, which results in progressive destruction of joint and bone tissues
This phase of arthritis has been extensively described in section 4 and 5, with the action of drugs - cytokine inhibitors. Nevertheless, the involvement of cytokines in the previous stages of inflammation before destruction has not been described, which is another inconsistency in the manuscript.
Also, in my opinion, the description of the action of drugs in RA in relation to cytokines needs to be reorganized as well.
The authors cite many articles (137), about 45 of which are older than 10 years. In my opinion the readers expect rather newer information. I suggest selecting a new date and description of the subject in relation to the latest reports and omission of historical data.
Conclusions
The conclusions do not contain any new information.
In addition, RF and ACPA are mentioned in the conclusions, however there is no information
about them in the article.
Author Response
The introduction contains basic information about RA well-known and may be omitted.
>> According to the review’s advice, we omitted some part.
Figure 1.
The figure is not clear. I propose to change it so that it really shows the cytokine network, which is the topic of the article.
>>Thank you for your comments. We changed the picture, and legends.
The subsections are not clearly written. I suggest systematizing the descriptions, for example, by describing the source of cytokines and triggers, receptors (with localization), the effect of activation, and summarizing the effects on key joint cells. Then, add a reference to current and future therapies.
>> Thank you for the advice. We rewrote the text according to reviewer’s comment.
★Lines 70-104 describe the action of TNF receptors, but without reference to RA. I have not found any information on which cells are these receptors and how their activation relates to arthritis.
>>Thank you for your comments. We added sentences after your advices.
Line 91 of revised manuscript -
“TNFR1 is expressing ubiquitously, whereas TNFR2 is expressing on T cells, myeloid cells, and endothelial cells. Upon activation, a precisely controlled multistep ubiquitination process activates NF-kB[14]”
Line 113 of revised manuscript -
“When TNFR1 is knocked out in collagen-induced arthritis (CIA) model, disease activity decreased, whereas TNFR2 knockout mice developed severe arthritis. Furthermore, both TNFR1 antagonist and TNFR2 agonist ameliorate severity of arthritis in CIA model[26, 27, 28, 29]. When stimulated with TNFR2 specific agonist, proportion of Treg cells was increased, and the histological score of arthritis also was ammeriolated in CIA mouse[27]. Thus TNFR1 signal seems to function to activate inflammatory target genes, whereas TNFR2 signal seems to func-tion as protective by regulating T regulatory (Treg) cells function.”
Line 137 of revised manuscript -
“Although precise signaling mechanism through TNFR2 in Treg is under investigation, increased proliferation and prolonged survival induced by TNFR2 activation is impaired in RelA deficient Treg cells.[34, 35]. Recently, another ligand of TNFR2, progranulin (PGRN) is identified. PGRN bind TNFR2 600-fold higher binding affinity than TNF[36]. Stimulated TNFR2 by PGRN interact with 14-3-3ε in their intracellular domain, and activated downstream cascade through PI3K/Akt/mTOR, leading to restrict NF-κB activation while simultaneously stimulating C/EBPβ activation[37] Knockout of 14-3-3ε or PGRN exacerbate arthritis in CIA model, with increase of proinflammatory macrophage and decrease of Treg cells in affected joints.”
★Line 107 “the cellular source of TNF and its relationship to the complex pathogenesis of RA has remained unclear” I suggest looking for information on this topic.
>>Thank you for your comments. We added the information about this, and changed sentences from line 104 of revised manuscript.
“Because of its multidirectional function and variety of cellular source as mentioned above, TNF from which cellular source assume respective pathogenic process remain unclear.”
★Verses 135-169 describe epigenetic mechanisms. The entire section lacks information on how TNF is involved in the pathogenesis of subsequent stages of arthritis in RA (topic of article).
>>Thank you for your comments. We added the information about RA pathogenesis, and added sentences.
Line95 of revised manuscript –
TNF is one of the key regulators of RA pathogenesis. Its expression is increased in RA patients, and overexpression of TNF caused autoimmune arthritis in transgenic animals[1, 18]. TNF signaling is involved in pathogenic process of RA multidirectionally. It activates endothelial cells and recruit proinflammatory cells, and synovial fibroblasts and macrophage to release proinflammatory cytokines, such as IL-6, IL-1β, and TNF[14,18, 21, 22]. It also controls the development of Th1 and Th17 T cells development, antibody production, and osteoclast differentiation[11, 23, 24]. We discuss about recent advances in TNF function in rheumatoid arthritis, especially focusing on its cellular source and contribution to the pathogenic process of RA, function of TNFRs, Treg development, and epigenomics.
Line 164 of revised manuscript -
The levels of CXCL8 in synovial fluid or peripheral blood is higher in the RA patients than in the healthy control. CXCL8 cause migration of immune cells to the joints, leading to joint destruction[43]. CXCL10 accelerated the migration of inflammatory cells through CXCR3 dependent manner, and induce RANKL expression in CD4+ Tcells. Cxcl10–/– and Cxcr3–/– mice with CAIA showed milder joint destruction than wild type[44], and blood level correlate RA disease activity[45]
Line 188 of revised manuscript -
Since then, several blocking agents were approaved with favorable clinical efficacy, and widely used in daily clinical practice. [47]. However, current anti-TNF drugs inhibit not only pathogenic TNF but also inhibit protective TNF derived from T cells that protect CIA development by controlling Th1 function [25]. It also inhibits TNFR2, that protect inflammation through Treg function as described above[14, 29 , 32 , 33, 34, 35]. Thus blocking TNF raise the risk of inhibiting the activity of some suppressor cells, and we sometimes experienced exacerbation of autoimmune disease such as psoriasis, lupus-like syndrome, multiple sclerosis, and sarcoidosis during the TNF treatment [48]. Reagents such as specific inhibitor of TNFR1 could fix these problems [49 ]
Rheumatoid arthritis is mainly associated with chronic inflammation, which results in progressive destruction of joint and bone tissues. This phase of arthritis has been extensively described in section 4 and 5, with the action of drugs - cytokine inhibitors. Nevertheless, the involvement of cytokines in the previous stages of inflammation before destruction has not been described, which is another inconsistency in the manuscript.
>> Thank you so much for the precious comment. We rewrote the detailed the involvement of cytokines in the previous stages of inflammation before bone and joint destruction in Chapter 3, by describing how IL23/IL-17 signaling pathway involves the previous stages of inflammation. The corrected part is as follows;
“IL-23 is a member of the IL-12 cytokine family composed of the IL-23 p19 subunit and the IL-12/23 p40 subunit. It is secreted by activated macrophage and dendritic cells in peripheral tissues such as skin, intestinal mucosa, joints, and lungs [68,69].
IL-23 mainly induces the differentiation of αβ T CD4+ naïve cells (Th0 cells) in T helper type 17 (Th17) cells [68] and stimulates the production of proinflammatory cytokines such as TNF-α, IL-1β, IL-21, and IL-17 from Th17 cells, and IL-6 from macrophages and dendritic cells [70]. γδT cells and innate lymphoid cells constitutively express the IL-23 receptor (IL-23R). IL-23R is a heterodimeric receptor composed of 2 subunits: IL-12Rβ1 and IL-23Rα.
The latter is specific to IL-23 signaling [68]. IL-23Rα interacts with JAK2, including STAT3 phosphorylation and leads to upregulation of retinoid-related orphan receptor gamma tau (RORγτ) and results in the development of Th17 cells [71].
IL-17 is involved in both early and established RA disease. It promotes activation of FLS, osteoclastogenesis, recruitment and activation of neutrophils, macrophages and B cells [72]. IL-17A is the first described member of IL-17 cytokine family. IL-17 cytokine family included 6 members, IL-17A to IL-17F [73]. TGF-β, IL-6, and IL-21 activate T lymphocytes and promote the initial differentiation of Th0 into Th17 cells and give responsiveness to IL-23 [68]. These cellular events are crucial step for Th17 cells stabilization and expansion.
IL-17 receptor (IL-17R) is expressed on various cell types such as epithelial cells, B and T cells, fibroblasts, monocytic cells, and bone marrow stroma [74]. Main roles of IL-17 include the promotion and the initiation of chemotaxis and the recruitment and activation of neutrophils in inflamed tissues. In inflammatory conditions such as inflammatory bowel diseases and arthritis, serum and tissue levels of IL-17 are increased, comparing with a non-pathological setting where IL-17A levels are extremely low or undetectable in human sera [68].
IL-7 is expressed by stromal cells in primary lymphoid organs and is known for the critical role in the development and homeostatic expansion of T cells in humans and mice [75]. IL-7 is overexpressed in inflamed tissues of patients with rheumatic autoimmune diseases and the expression levels of IL-7 is associated with clinical parameters of disease [76]. The differentiation of Th17 cells is mainly mediated by STAT3 signaling through cytokines such as IL-6, IL-21 and IL-23. In the phase of T-cell activation and differentiation, IL-7 receptor (IL-7R) is expressed on Th17 cells. In the differentiation stage, IL-7R is re-expressed in activated Th17 cells and IL-7 is critically needed to sustain survival and expansion of differentiated Th17 cells through STAT5 signaling [77].
Several T-cell subsets and their complex interactions likely contribute to RA pathology. It is largely accepted that regulatory T cells accumulate in RA synovial fluid and that the equilibrium between them and effector cells is a key factor in controlling the inflammatory processes involved in RA [78]. Th17 cells are known to comprise a third T-cell subset since Th1/Th2 cells, and are induced by the cytokines IL-6, IL-1β, IL-21, TGF-β, and IL-23. They are present in synovial joints [79] and secrete IL-17, which is a pro-inflammatory cytokine contributing to osteoclastogenesis along with TNF and IL-6.
Synergism between IL-17 and TNF-α has been shown to activate the production of pro-inflammatory mediators such as IL-1β, IL-6, IL-8, prostaglandin E2, and MMPs. Then it promotes progression of early inflammation toward a chronic arthritis [68].
Recently, the proportion of receptor CCR6+ Th17 cells is reported to be increased in the peripheral blood of treatment-naïve patients with early RA [80]. In addition, higher frequencies of Th17 cells have been observed in the synovium of RA patients compared to OA patients [81].
The differentiation of osteoclast is induced significantly in the presence of IL-17 either directly or indirectly through upregulation of RANKL [68].
Bone erosion by osteoclasts is one of the most important pathologic features of RA [82,83]. Receptor activator of nuclear factor kappa-B ligand (RANKL) is a cytokine belonging to the TNF superfamily and mainly stimulates osteoclast differentiation. It binds to receptor activator of nuclear factor kappa-B (RANK) on osteoclasts and osteoclast precursors and promotes osteoclast differentiation and activation. Osteoprotegerin (OPG) is a soluble decoy receptor of RANKL, and upon binding to RANKL inhibits the activation of RANK. The inflamed synovium and pannus in RA produce significantly higher levels of RANKL and lower levels of OPG in comparison to healthy synovium [84,85]. The cells responsible for the increased expression of RANKL in the inflamed synovial membrane are fibroblast-like synoviocytes (FLS) and T lymphocytes. The increased RANKL/OPG ratio promotes osteoclast differentiation and activation at the synovium-bone interface and the development of bone erosions in RA [84,85].
RANKL on T cells or fibroblasts, which are activated by a combination of IL-6, IL-17, and TNF, regulates the differentiation of osteoclasts. IL-6 signaling induces RANKL expression in RA-FLS through expression of NFATc1 and TRAP5b mRNA in co-cultures of RA-FLS and osteoclast precursor cells [86].
IL-17 increases RANKL expression in adjuvant-induced arthritis-derived synovial fibroblasts, leading to increased osteoclastogenesis in vitro. As both RANKL expression and osteoclastogenesis are reduced by blocking IL-17R and STAT-3, they are dependent on these molecules [87]. Furthermore, IL-17 modulates osteoclast precursor cells. Raw264.7 cell culture experiments have shown that a low level of IL-17A promotes the RANKL-RANK system by mediating the JNK signaling pathway and activating autophagy and osteoclastogenesis in induced osteoclast precursor cells. However, a high level of IL-17A inhibits osteoclastogenesis [88]. A RANKL-independent osteoclast differentiation pathway has also been reported.
TNF induces differentiation of osteoclasts from mouse bone marrow myeloid cells (mBMM) and human peripheral blood monocytes (PBMC) independently of RANKL function [89]. TNF receptor associated factor-3 (TRAF3) limits RANKL-induced osteoclast formation by promoting proteasomal degradation of NF-kB-inducing kinase in a complex with TRAF2 and cIAP. It also inhibits osteoclast formation induced by TNF. Hydroxychloroquine is anti-inflammatory drugs for rheumatoid arthritis and prevents TRAF3 degradation in osteoclast precursors and inhibits osteoclast formation in vitro [90].
Tartrate-resistant acid phosphatase (TRAP)-positive cells induced by TNF lack bone absorption ability, but gain it upon treatment with a combination of IL-6 and TNF. Thus IL-6 is required for the bone-resorbing activity of TRAP-positive cells induced by TNF. Interestingly, whereas bone absorption activity cannot be abolished in mice with STAT3 conditional knockout, treatment with JAK inhibitor or MEK inhibitor achieves this [91]. Furthermore, the number of osteoclasts induced ex vivo from RA patient PBMC with TNF and IL-6 is positively correlated with the host patient modified total Sharp score [92].
IL-17 A also promotes MMP production in chondrocytes [93].
Antibodies against IL-17 (ixekizumab and secukinumab) or IL-17R (brodalumab) have been examined in patients with RA [94- 97]. In a phase I RCT of RA patients treated with oral DMARDs, the addition of ixekizumab improved RA signs and symptoms and DAS28, compared to placebo [94].
This improvement was confirmed in a phase II study in which Ixekizumab was administered to patients who were naïve to biological therapy or resistant to TNF-a inhibitors [98]. In a phase II study enrolling RA patients with inadequate response to methotrexate, greater decreases in DAS28 were observed with Secukinumab than with placebo [95]. ”
Section 3 describes the cellular structure of the joint synovium and does not address the subject of the manuscript. I suggest changing the section to relate to the topic.
>>We delete this section, according to your advice.
The authors cite many articles (137), about 45 of which are older than 10 years. In my opinion the readers expect rather newer information. I suggest selecting a new date and description of the subject in relation to the latest reports and omission of historical data.
>> According to the review’s advice, we cited newer information.
Conclusions. The conclusions do not contain any new information. In addition, RF and ACPA are mentioned in the conclusions, however there is no information. about them in the article.
>> According to the review’s advice, we rewrote conclusion with RF and ACPA.
Round 2
Reviewer 1 Report
A few typing errors need to be revised
No any further comment.
Reviewer 2 Report
Regarding the Authors' revisions- the manuscript is worth publishing. The authors have satisfactorily addressed all my concerns. Thank you for your work.
Reviewer 3 Report
I maintain my first opinion of the article.
The authors have made changes but the quality of the article has not improve.
I usually expect authors to discuss my comments, but I do not approve of accepting the comment in the response without modifying the manuscript accordingly.
Despite the depth of the manuscript, the conclusions do not extend beyond the basic knowledge of the subject.
In my opinion, the article does not meet the high requirements of the journal.